# Masked Multi-path Contrast with Confidence-Gated Semantic Imputation for Incomplete Multi-view Clustering

**Fan Yang** [* 1]  **Haikun Xu** [* 1]

## Abstract

Incomplete multi-view clustering (IMVC) becomes particularly challenging under heavy missingness and view-availability imbalance. In this regime, scarce co-observed pairs make cross-view correspondences unreliable. Imputation-first pipelines may trigger cascading reconstruction errors, while purely consistency-based alignment often degrades sharply and gives limited control over semantic convergence across views. We propose *MAGIC* (Masked multi-pAth contrast with confIdence-Gated semantIc imputation), a unified framework that learns calibrated cluster semantics before conservative completion. MAGIC builds multiple correlated representation and prediction paths from lightly augmented latent codes, and couples them with a masked multi-path contrastive consensus objective and prediction-consistency regularization. The resulting posteriors are aggregated into view-wise soft assignments to reduce overconfidence and alleviate dominance by more frequently observed views. Based on these calibrated semantics, MAGIC performs similarity-guided semantic transfer in label space with confidence-aware gating, and completes missing representations through a geometry-preserving prototype fallback. Experiments on four benchmarks across different missing ratios show consistent gains over prior IMVC methods, and the ablations support the roles of masked multi-path consensus learning and confidence-gated semantic imputation.

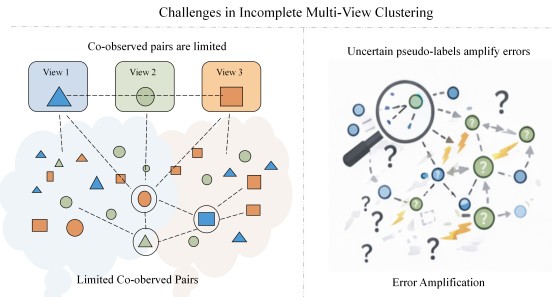

*Figure 1.* Motivation of two dominant failure modes under high missingness. (Left) Co-observed cross-view pairs become scarce, which makes alignment fragile and biases optimization toward frequently observed views. (Right) Unreliable pseudo-labels can be propagated through training and imputation, which gradually amplifies errors and weakens clustering. These two effects motivate consensus-driven learning with confidence-gated semantic transfer.

## 1. Introduction

Multi-view clustering (MVC) (Moujahid & Dornaika, 2025; Chowdhury et al., 2025; Qin et al., 2025; Zhao et al., 2025; Hu et al., 2022) exploits complementary information across heterogeneous sources and often recovers latent structure more reliably than single-view learning. In real-world deployments, however, sensor faults, occlusions, and transmission loss frequently lead to missing views, making the fully observed assumption unrealistic. This motivates incomplete multi-view clustering (IMVC) (Zhao et al., 2024; Dong et al., 2025a; Yuan et al., 2025c), where deep models are increasingly adopted for their representation capacity and empirical performance.

Recent deep IMVC approaches (Wang et al., 2024; Chao et al., 2024a; Liu et al., 2024b; Zheng & Tang, 2024; Du et al., 2026) largely follow two lines. The first performs view completion, explicitly or implicitly, before clustering, for example by exploiting low-rank inter-view relations (Zhao et al., 2023; Ren et al., 2024) or conducting explicit completion prior to optimization (Huang et al., 2024; Yuan et al., 2025b). While completion can reduce information loss, it is prone to error accumulation when missingness is high. The second line reduces reliance on imputation by enforcing

---

[*]Equal contribution  [1]School of Computer Science and Artificial Intelligence, Nanjing University of Finance and Economics, Nanjing, Jiangsu, China. Correspondence to: Fan Yang <nufe_yf@163.com>.

*Proceedings of the 43rd International Conference on Machine Learning*, Seoul, South Korea. PMLR 306, 2026. Copyright 2026 by the author(s).

cross-view consistency in a shared latent space, including distillation-based schemes (Wu et al., 2025) and prototypical or semantic contrastive learning (Li et al., 2023; Sun et al., 2024). These methods can be more robust, yet they still degrade markedly when co-observed samples are scarce and view availability is highly imbalanced.

This paper focuses on the high-missingness regime and highlights the two dominant failure modes in Fig. 1. First, scarce co-observations weaken alignment signals and make optimization vulnerable to view dominance, leading to noisy cross-view correspondence learning. Second, assignment uncertainty can trigger error propagation: early pseudo-labels may be overconfident yet incorrect, biasing representation learning and contaminating subsequent completion or alignment. Together, these effects destabilize semantic convergence across views and limit robustness in the most challenging settings. We note that some recent IMVC studies aim to learn under extremely weak, unpaired, or even absent co-observation by exploiting structural or semantic invariance (Xin et al., 2025; Dong et al., 2026). Our focus is different: rather than treating the lack of co-observed pairs as the only bottleneck, we study the coupled risks of sparse alignment signals, biased view availability, and unreliable semantic propagation. MAGIC is therefore designed to stabilize posterior semantics before performing conservative transfer.

To address these issues, we propose MAGIC (Masked multi-pAth contrast with confIdence-Gated semantIc imputation for incomplete multi-view Clustering), a unified framework for severe missingness. MAGIC has two main components. The first is multi-path contrastive consensus, which jointly optimizes fused, per-view, and masked-fusion paths and encourages agreement at both representation and prediction levels. This design reduces dependence on scarce paired samples, mitigates view dominance, and produces more reliable posteriors for clustering. The second is confidence-gated semantic imputation, which transfers class-level semantics only when the source evidence is sufficiently reliable. This conservative transfer reduces the spread of incorrect assignments and keeps cluster semantics more stable during completion.

Our contributions are summarized as follows:

- We propose MAGIC, a deep IMVC framework for high-missingness settings, which combines multi-path consensus learning with confidence-gated semantic imputation.

- We formulate a multi-path contrastive consensus objective over fused, per-view, and masked-fusion paths. This objective improves posterior stability when co-observed samples are limited and reduces the influence of more frequently observed views.

- We introduce a confidence-gated, class-level semantic imputation mechanism. It transfers semantics only under reliable evidence and uses prototype-based completion for missing views, which reduces error propagation under severe missingness.

- We compare MAGIC with recent IMVC baselines on four benchmarks with different data scales, view configurations, and missing ratios. The results show consistent gains across the evaluated settings.

## 2. Related Work

**Incomplete MVC.** Related work on IMVC has evolved from explicit view completion to structured recovery and contrastive consistency learning. Early methods (Zhao et al., 2023; Huang et al., 2024; Yuan et al., 2025b; Xing et al., 2024; Zheng & Tang, 2024; Li et al., 2024; Du et al., 2026) complete missing data or affinity structures, such as raw features, graphs, or hierarchical semantics, and couple completion with clustering in a joint objective. To improve robustness and reduce computational cost, subsequent approaches (Long et al., 2024; Deng et al., 2024; Zhang et al., 2023; Zheng, 2025) move recovery into feature subspaces, spectral embeddings, or tensor-regularized representations, using low-rank priors and shared projections to enable implicit completion. Building on structural assumptions, geometric, graph, and tensor formulations (Yang et al., 2024; Chao et al., 2024b; Wang et al., 2025; Xu et al., 2019; Dong et al., 2024; 2025c) further impose cross-view consistency or generative alignment to preserve topology and mitigate error accumulation. More recent work emphasizes robustness and separability through prototype- or semantic-level contrast with consistency regularization (Lin et al., 2021; 2022; Wen et al., 2020; Li et al., 2023; Sun et al., 2024; Lin et al., 2023; Dai et al., 2025; Hu et al., 2025), and refines positive/negative construction or applies selective alignment to reduce false negatives and over-alignment under missing views (Yu et al., 2025; Ding et al., 2025; Li et al., 2025b; Dong et al., 2025b). To handle complex structure and larger scale, methods in (Xue et al., 2025; Xin et al., 2025; Zhang et al., 2025b; Yin et al., 2025; Dong et al., 2025d) combine multi-graph or multi-level contrast with limited-pair generative alignment, view-graph fusion, and latent reconstruction. I2MVC (Wu et al., 2025) follows a different route and reduces reliance on explicit imputation through knowledge distillation. Related incomplete multi-view learning beyond clustering has also studied noisy labels and incomplete views (Li et al., 2025a), but MAGIC focuses on fully unsupervised clustering. Despite these advances, high missingness with strong view imbalance remains challenging: alignment signals are sparse and uneven, pseudo-label uncertainty is amplified, and cluster semantics can drift without explicit stabilization. This motivates approaches that

strengthen consensus across heterogeneous learning paths while controlling the reliability of semantic transfer under missing views.

**Contrastive Learning for IMVC.** Contrastive learning has become a primary mechanism for exploiting cross-view agreement in incomplete multi-view clustering. Early deep IMVC methods mainly use instance-level contrast, treating augmented or cross-view representations of the same sample as positives and other samples as negatives, as in ICMVC (Chao et al., 2024b) and I2MVC (Wu et al., 2025). However, instance-only objectives can be fragile when missingness and view noise corrupt the positive/negative sets, leading to false negatives and unstable optimization. Recent studies therefore incorporate semantic structure into contrastive learning: prototype-, pseudo-label-, and label-aware objectives (Li et al., 2023; Sun et al., 2024; Lin et al., 2023; 2021; 2022; Wen et al., 2020; Dai et al., 2025; Hu et al., 2025) encourage agreement at a higher granularity, typically improving separability and reducing sensitivity to sparse or unreliable correspondences. Meanwhile, scarce co-observed pairs and non-uniform missing patterns motivate selective alignment; methods such as (Zhang et al., 2025b; Yin et al., 2025; Ding et al., 2025; Dong et al., 2025b) refine pair construction or restrict alignment to reliable pairs to reduce erroneous matching and avoid over-alignment to dominant views. Complementarily, uncertainty-aware designs introduce prediction-level signals: prediction-driven constraints or consistency regularization in clustering spaces (Yu et al., 2025; Ding et al., 2025) align posterior distributions or assignments rather than raw features, which can be more robust under partial observability.

## 3. Method

### 3.1. Problem Setup

We study incomplete multi-view clustering with inputs $\{X_v\}_{v=1}^V$, where $v \in \{1, \dots, V\}$ indexes the view and each $X_v$ contains $N$ instances. Missingness is encoded by a binary visibility matrix $G \in \{0, 1\}^{N \times V}$, where $G(i, v) = 1$ if instance $i$ is observed in view $v$ and $G(i, v) = 0$ otherwise. The goal is to partition the $N$ instances into $K$ clusters. MAGIC combines two mechanisms to address high missingness: (i) multi-path contrastive consensus learning, which stabilizes representation learning when co-observed cross-view pairs are scarce; and (ii) confidence-gated semantic imputation, which completes missing semantics conservatively to reduce error propagation.

### 3.2. Multi-path Contrastive Consensus Learning

This module addresses fragile alignment under missingness. Direct cross-view matching can be noisy and can be biased toward more frequently observed views. To reduce

this dependence, MAGIC constructs three correlated paths from the same incomplete instances, namely a fused path, a per-view path, and a masked-fusion path. It then enforces agreement across these paths in both representation space and prediction space.

For each view $v \in \{1, \dots, V\}$, we use a view-specific autoencoder $(E_v, D_v)$. Let $x_v^i$ denote the observation of instance $i \in \{1, \dots, N\}$ in view $v$, and let $\hat{x}_v^i$ be its reconstruction. The latent codes are $Z_v = E_v(X_v)$, and reconstruction is optimized only over observed entries:

$$L_{\text{rec}} = \frac{1}{|\Omega|} \sum_{v=1}^V \sum_{i=1}^N G(i, v) \, \|x_v^i - \hat{x}_v^i\|_2^2, \qquad (1)$$

where $\Omega = \{(i, v) \mid G(i, v) = 1\}$ is the set of observed view–instance pairs, $|\Omega|$ is its cardinality, and $\|\cdot\|_2$ is the Euclidean norm.

To obtain robust learning signals without requiring dense co-observation across views, we operate in latent space. For each view $v$, we apply two independent stochastic augmentations and obtain $Z_v^{(1)}$ and $Z_v^{(2)}$. From $\{Z_v^{(a)}\}_{v=1}^V$ with $a \in \{1, 2\}$, we construct: (i) fused representations $(H^{(1)}, H^{(2)})$ by aggregating the available views to capture shared semantics; (ii) per-view representations $\{(H_v^{(1)}, H_v^{(2)})\}_{v=1}^V$ that preserve view-specific cues while remaining tethered to fused semantics; and (iii) masked-fusion representations $(H_m^{(1)}, H_m^{(2)})$ by randomly dropping a fixed fraction of latent dimensions before fusion to improve robustness to partial corruption. For any contrastive pair of sources $(A, B)$, we only evaluate losses on the shared index set $\mathcal{I}(A, B)$ where both sources are defined, so that missing entries never create invalid positive pairs.

We measure dependence between two embedding sets with InfoNCE (Oord et al., 2018). Let $A = \{a_i\}_{i \in \mathcal{I}}$ and $B = \{b_i\}_{i \in \mathcal{I}}$ be $\ell_2$-normalized embeddings sharing the index set $\mathcal{I} = \mathcal{I}(A, B)$. With cosine similarity $\text{sim}(a, b) = a^\top b$ and temperature $\tau > 0$, we use

$$\ell(A, B) = -\frac{1}{|\mathcal{I}|} \sum_{i \in \mathcal{I}} \log \frac{\exp(\text{sim}(a_i, b_i)/\tau)}{\sum_{j \in \mathcal{I}} \exp(\text{sim}(a_i, b_j)/\tau)}. \quad (2)$$

It is shown in (Oord et al., 2018) that the population InfoNCE objective lower-bounds mutual information; we state the formal result in Appendix A (Proposition A.1).

We couple the three paths using nonnegative weights $\alpha_f, \alpha_p, \alpha_m \geq 0$:

$$L_{\text{mpc}} = \alpha_f \, \ell\big(H^{(1)}, H^{(2)}\big)$$
$$+ \frac{\alpha_p}{2} \sum_{v=1}^V \big(\ell\big(H_v^{(1)}, H^{(1)}\big) + \ell\big(H_v^{(2)}, H^{(2)}\big)\big) \quad (3)$$
$$+ \frac{\alpha_m}{2} \big(\ell\big(H_m^{(1)}, H^{(2)}\big) + \ell\big(H_m^{(2)}, H^{(1)}\big)\big).$$

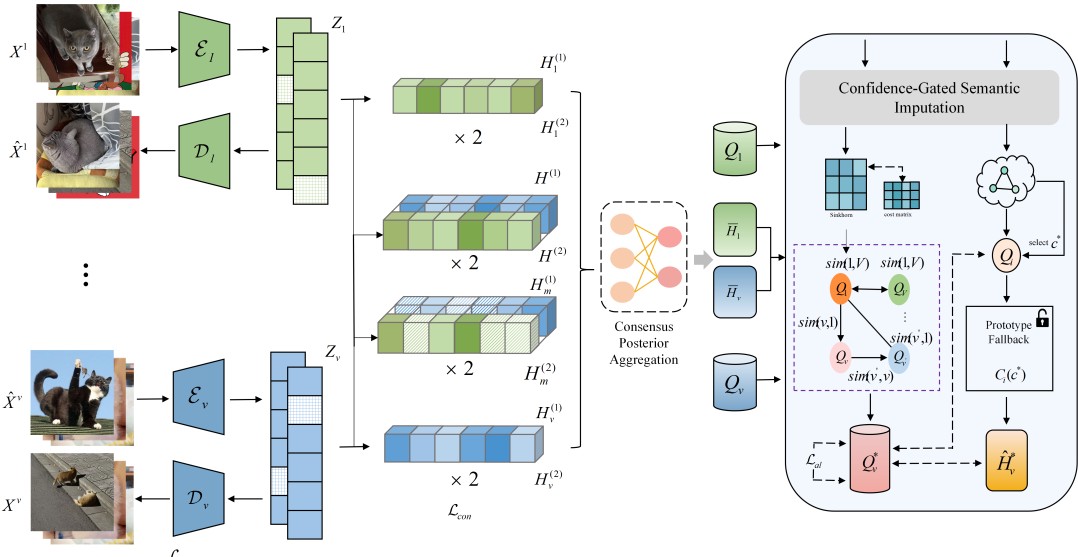

*Figure 2.* Overview of MAGIC. MAGIC learns view-specific representations with reconstruction and multi-path contrastive consensus, then aggregates view-wise predictions into a calibrated consensus posterior. The resulting posteriors guide confidence-gated semantic imputation, which transfers cluster semantics through optimal-transport prototype matching and completes missing views via prototype fallback, enabling robust IMVC under high missingness and view imbalance.

The fused term stabilizes global semantics across augmentations; the per-view term prevents any single view from dominating by aligning each view to fused semantics; and the masked-fusion term encourages invariance to partial feature corruption, which is critical when missingness makes alignment intermittent.

To reduce augmentation-specific overconfidence, we add a prediction-consistency regularizer that penalizes discrepancies between matched posteriors produced by the same path under two augmentations. For categorical distributions $p, q \in \Delta^{K-1}$ over $K$ clusters, where $\Delta^{K-1} = \{r \in \mathbb{R}_{\geq 0}^K : \sum_{k=1}^K r(k) = 1\}$, define the symmetric KL divergence

$$\mathrm{SKL}(p, q) = \mathrm{KL}(p\|q) + \mathrm{KL}(q\|p),$$

with KL the Kullback–Leibler divergence. We then define

$$L_{\mathrm{pc}} = \frac{1}{|\mathcal{J}|} \sum_{i \in \mathcal{J}} \mathrm{SKL}\Big(P_i^{(1)}, P_i^{(2)}\Big),$$

where $P_i^{(1)}$ and $P_i^{(2)}$ denote the matched posteriors for instance $i$ under the two augmentations, and $\mathcal{J}$ is the valid index set of the corresponding path pair. For fused and masked-fusion paths, this set contains all samples in the mini-batch after aggregation from available views; for a per-view path, it contains only the samples where that view is observed. The final objective in this module is

$$L_{\mathrm{con}} = L_{\mathrm{mpc}} + \lambda_{\mathrm{pc}} \, L_{\mathrm{pc}}, \tag{4}$$

where $\lambda_{\mathrm{pc}} \geq 0$ controls the strength of prediction consistency. The calibrated posteriors are then aggregated and used for conservative imputation.

### 3.3. Consensus Posterior Aggregation

The consensus module yields multiple correlated posteriors, and relying on a single path can be brittle early in training under view imbalance. We therefore aggregate path-wise posteriors into a single consensus posterior per view.

For view $v \in \{1, \ldots, V\}$, let $P_f$ be the fused-path posterior, $P_p^v$ the per-view-path posterior, and $P_m$ the masked-fusion posterior, where each is a categorical distribution in $\Delta^{K-1}$. Since the fused and masked-fusion posteriors do not depend on $v$, we write $P_f^v := P_f$ and $P_m^v := P_m$, and let $P_r^v$ denote the posterior for path index $r \in \{f, p, m\}$. With normalized weights $\pi_r \geq 0$ satisfying $\sum_{r \in \{f,p,m\}} \pi_r = 1$ and aggregation temperature $\tau_{\mathrm{agg}} > 0$, we form a temperature-scaled log-opinion pool

$$Q^v(k) = \frac{\exp\left(\tau_{\mathrm{agg}}^{-1} \sum_{r \in \{f,p,m\}} \pi_r \log(P_r^v(k) + \epsilon)\right)}{\sum_{t=1}^K \exp\left(\tau_{\mathrm{agg}}^{-1} \sum_{r \in \{f,p,m\}} \pi_r \log(P_r^v(t) + \epsilon)\right)}, \tag{5}$$

where $\epsilon$ is a small numerical constant. This aggregation reduces the influence of isolated overconfident predictions and keeps signals that are supported by multiple paths. In the main experiments, we use fixed coefficients $(\pi_f, \pi_p, \pi_m) = (1/3, 1/6, 1/2)$ and $\tau_{\mathrm{agg}} = 0.8$ for all

datasets. The resulting $\{Q^v\}_{v=1}^V$ are used as calibrated soft labels for confidence-gated imputation.

For clarity, the posteriors used in MAGIC have different roles. The path-wise posteriors $P_f$, $P_p^v$, and $P_m$ are produced by the fused, per-view, and masked-fusion paths before aggregation. The view-wise consensus posterior $Q^v$ is the calibrated stage-1 semantic output for view $v$. During stage-2 imputation, $Q^v$ is sharpened into $\tilde{q}^v$ to compute class masses, class prototypes, and view similarities. After confidence-gated transfer from the selected source view, $\hat{q}_n^v$ denotes the post-transfer target-view posterior for instance $n$, which is used to determine the prototype fallback for missing-view completion.

### 3.4. Confidence-Gated Semantic Imputation

This module addresses error propagation in missing views by transferring semantics at the class level and activating transfer only when the source is confident. We use $n \in \{1, \ldots, N\}$ for instances and $v \in \{1, \ldots, V\}$ for a target view. Let $m_n^v := G(n, v) \in \{0, 1\}$ indicate visibility, and let $h_n^v \in \mathbb{R}^d$ be the latent feature in view $v$, where $d$ is the latent dimension. We sharpen the consensus posterior $Q^v$ into $\tilde{q}^v \in \Delta^{K-1}$ (sharpening is defined in Appendix B) and compute class prototypes and class masses using only observed samples:

$$C_k^v = \frac{\sum_{n=1}^N m_n^v \tilde{q}_{nk}^v h_n^v}{\sum_{n=1}^N m_n^v \tilde{q}_{nk}^v + \epsilon}, \qquad a_k^v = \frac{\sum_{n=1}^N m_n^v \tilde{q}_{nk}^v + \epsilon}{\sum_{t=1}^K \sum_{n=1}^N m_n^v \tilde{q}_{nt}^v + K\epsilon},$$

$$(6)$$

where $C_k^v \in \mathbb{R}^d$ is the prototype of cluster $k$ in view $v$, and $a^v = (a_1^v, \ldots, a_K^v)$ is a normalized nonnegative class-mass vector. In implementation, prototypes are not learnable parameters and are not updated by momentum. They are recomputed online in each forward pass from the current mini-batch by masking out missing samples and taking soft-assignment-weighted centroids over currently observed samples, with a small constant used for numerical stability.

We estimate view similarity $s_{vu} \in [0, 1]$ in label space using only co-observed instances. Let $\mathcal{O}_{vu} = \{n \mid G(n, v) = 1, G(n, u) = 1\}$ denote the co-observed set of views $v$ and $u$. We compute

$$s_{vu} = \frac{1}{|\mathcal{O}_{vu}|} \sum_{n \in \mathcal{O}_{vu}} \frac{\langle \tilde{q}_n^v, \tilde{q}_n^u \rangle}{\|\tilde{q}_n^v\|_2 \|\tilde{q}_n^u\|_2}, \tag{7}$$

and set $s_{vu} = 0$ if $\mathcal{O}_{vu}$ is empty. The most compatible source view is selected by

$$u^\star = \arg\max_{u \neq v} s_{vu}. \tag{8}$$

Let $\langle \cdot, \cdot \rangle$ be the Euclidean inner product, and define cosine similarity $\cos(a, b) = \langle a, b \rangle / (\|a\|_2 \|b\|_2)$. The class-to-

class transport cost from $u^\star$ to $v$ is

$$\text{Cost}_{kk'}^{(v \leftarrow u^\star)} = 1 - \cos(C_k^v, C_{k'}^{u^\star}). \tag{9}$$

We compute an entropy-regularized coupling $\Pi^{(v \leftarrow u^\star)} \in \mathbb{R}_+^{K \times K}$ by

$$\Pi = \arg\min_{\Pi \geq 0} \langle \Pi, \text{Cost} \rangle + \varepsilon \sum_{k=1}^K \sum_{k'=1}^K \Pi_{kk'} (\log \Pi_{kk'} - 1)$$

$$(10)$$

$$\text{s.t.} \quad \Pi \mathbf{1} = a^v, \quad \Pi^\top \mathbf{1} = a^{u^\star},$$

where $\varepsilon > 0$ is the entropy weight, $\mathbf{1} \in \mathbb{R}^K$ is the all-ones vector, and $\langle \Pi, \text{Cost} \rangle = \sum_{k, k'} \Pi_{kk'} \text{Cost}_{kk'}$. Existence/structure properties for entropic OT and its Sinkhorn form are given in Appendix A (Proposition A.2).

**Theorem 3.1.** *Let $T \in \mathbb{R}_+^{K \times K}$ be row-stochastic, i.e., $T\mathbf{1} = \mathbf{1}$. For any $p, q \in \Delta^{K-1}$, the class-level transfer $p \mapsto pT$ is non-expansive in KL:*

$$\text{KL}(pT \| qT) \leq \text{KL}(p \| q). \tag{11}$$

*Consequently, it is also non-expansive for the symmetric KL, $\text{SKL}(pT, qT) \leq \text{SKL}(p, q)$.*

The proof follows from the data-processing inequality for stochastic maps and is provided in Appendix A. To gate semantic transfer, we use normalized entropy confidence. For a probability vector $q \in \Delta^{K-1}$, define its entropy $H(q) = -\sum_{k=1}^K q(k) \log q(k)$ (natural logarithm) and

$$\text{conf}(q) = 1 - \frac{H(q)}{\log K}. \tag{12}$$

Let $\tau_{\text{src}} \in [0, 1]$ be a confidence threshold. We convert $\Pi$ into a conditional class map $T^{(v \leftarrow u^\star)} \in \mathbb{R}_+^{K \times K}$ by setting $T_{k'k}^{(v \leftarrow u^\star)} = \Pi_{kk'} / a_{k'}^{u^\star}$ for all $k, k'$, so that each row of $T$ sums to 1 when $a_{k'}^{u^\star} > 0$. If $\text{conf}(q_n^{u^\star}) \geq \tau_{\text{src}}$, we transfer semantics by

$$\hat{q}_n^v = q_n^{u^\star} T^{(v \leftarrow u^\star)}, \qquad \hat{q}_n^v \leftarrow \frac{\hat{q}_n^v}{\|\hat{q}_n^v\|_1}, \tag{13}$$

where $q_n^{u^\star} \in \Delta^{K-1}$ is the source posterior for instance $n$, and $\|\cdot\|_1$ is the $\ell_1$ norm. We then define the predicted cluster index $c^\star(n) = \arg\max_{k \in \{1, \ldots, K\}} \hat{q}_n^v(k)$. For latent completion in the target view, we use a conservative prototype fallback

$$\hat{h}_n^v = C_{c^\star(n)}^v \qquad \text{when } m_n^v = 0. \tag{14}$$

An optional within-cluster $k$-nearest-neighbor refinement (including the distance and neighbor count) is provided in Appendix B; we keep the main text conservative to avoid injecting noisy feature-level signals when missingness is severe.

**Algorithm 1** Training Procedure of MAGIC

---

1: **Input:** $\{X_v\}_{v=1}^V$, $G$, $K$, epochs $T_0, T$, imputation start $T_{\mathrm{imp}}$, hyperparameters in Sec. 3.
2: **Output:** cluster labels $Y$.
3: Pretrain $\{(E_v, D_v)\}$ with masked reconstruction Eq. (1) for $T_0$ epochs.
4: **for** epoch $= 1$ to $T$ **do**
5:    Sample mini-batch; encode and build fused/per-view/masked-fusion paths (Sec. 3.2).
6:    Compute $L_{\mathrm{con}}$ via multi-path contrast Eq. (3) and prediction consistency Eq. (4).
7:    Aggregate path posteriors into $\{Q^v\}$ by Eq. (5).
8:    **if** epoch $\geq T_{\mathrm{imp}}$ **then**
9:       Run confidence-gated semantic imputation (Eqs. (6)–(13)) and latent completion Eq. (14).
10:   **end if**
11:   Update all parameters by minimizing the total loss Eq. (15).
12: **end for**
13: Inference: compute $\{Q^v\}$ for observed views, aggregate them by normalized-entropy confidence weights, and output $Y$ by the argmax of the aggregated posterior.

---

### 3.5. Overall Objective and Training

We optimize

$$L_{\mathrm{tot}} = L_{\mathrm{rec}} + \alpha\, L_{\mathrm{con}} + \omega\, L_{\mathrm{al}}, \qquad (15)$$

where $\alpha, \omega \geq 0$. The alignment term is a similarity-weighted symmetric KL regularizer over co-observed view pairs:

$$L_{\mathrm{al}} = \frac{1}{|\mathcal{P}|} \sum_{(v,u) \in \mathcal{P}} s_{vu} \frac{1}{|\mathcal{O}_{vu}|} \sum_{n \in \mathcal{O}_{vu}} \mathrm{SKL}(Q_n^v, Q_n^u), \quad (16)$$

where $\mathcal{P} = \{(v,u) \mid v < u, |\mathcal{O}_{vu}| > 0\}$ is the set of view pairs with co-observed samples. Thus, $L_{\mathrm{al}}$ aligns view-wise consensus posteriors only when both views are observed and gives larger weights to more semantically compatible view pairs.

The effective sample set is term-specific. The reconstruction loss is evaluated only on observed entries, and missing entries do not contribute reconstruction gradients. In the multi-path contrastive objective, the fused–fused term is computed on all mini-batch samples because the fused path is built from available views; the per-view–fused terms are computed only on samples where the corresponding view is observed; and the masked-fusion–fused terms are computed on all samples after masked fusion is constructed from available views. The prediction-consistency regularizer is applied to paired posteriors from the two augmentations for the fused path, the masked-fusion path, and each observed

per-view path. For cross-view alignment, the view-pair similarity is computed strictly on the co-observed subset of that view pair and then used to weight the corresponding alignment term. Thus, MAGIC does not backpropagate reconstruction or alignment signals through raw missing entries.

Training first optimizes (1)–(4) to obtain stable consensus posteriors, then enables confidence-gated imputation with a schedule on $\tau_{\mathrm{src}}$ (Appendix B). At inference, the final label is produced by confidence-weighted aggregation over the view-wise consensus posteriors rather than by a raw fused posterior or a fixed single-view posterior. For each observed view, we compute $\rho_n^v = \mathrm{conf}(Q_n^v)$, normalize these confidences as $\gamma_n^v = G(n,v)\rho_n^v / \sum_{u=1}^V G(n,u)\rho_n^u$, aggregate log-posteriors by $Q_n^{\mathrm{final}}(k) \propto \exp(\sum_v \gamma_n^v \log(Q_n^v(k) + \epsilon))$, and output $Y_n = \arg\max_k Q_n^{\mathrm{final}}(k)$. This rule is used for all reported results and naturally handles samples with only a subset of views observed.

## 4. Experiments

### 4.1. Datasets

We evaluate MAGIC on four public IMVC benchmarks covering diverse domains, view configurations, and dataset scales. FMNIST (He et al., 2025) contains 60,000 instances from 10 categories. We use three pre-extracted views with feature dimensions 342, 1024, and 64, respectively. Fashion (Moujahid & Dornaika, 2025) is a three-view benchmark with 30,000 samples, where each instance is represented by three images from the same category. BDGP (Liu et al., 2024a) includes 2,500 samples from 5 classes with two complementary views. MNIST-USPS (Shu et al., 2024) contains 5,000 paired handwritten digits over 10 classes, where the two views come from the MNIST and USPS domains. Following common IMVC protocols, we simulate incompleteness by sampling missing views under different missing ratios $\eta$.

### 4.2. Baselines and Experimental Setups

To validate the effectiveness of our approach, we benchmark MAGIC against ten state-of-the-art multi-view clustering baselines, including COMPLETER (Lin et al., 2021), DSIMVC (Tang & Liu, 2022), APADC (Xu et al., 2023), CPSPAN (Jin et al., 2023), ProImp (Li et al., 2023), DIVIDE (Lu et al., 2024), GIMVC (Bai et al., 2024), DCG (Zhang et al., 2025a), PMIMC (Yuan et al., 2025a), and ROLL (Sun et al., 2025). To comprehensively assess performance under missing views, we synthesize incomplete multi-view data by sampling a binary visibility matrix for each dataset while enforcing that every instance retains at least one visible view. The missing ratio $\eta$ is varied over $\{0.1, 0.3, 0.5, 0.7\}$.

*Table 1.* Clustering performance (%) under different missing ratios $\eta$.

| Dataset | Method | References | $\eta = 0.1$ | | | $\eta = 0.3$ | | | $\eta = 0.5$ | | | $\eta = 0.7$ | | |
|---|---|---|---|---|---|---|---|---|---|---|---|---|---|---|
| | | | ACC ↑ | NMI ↑ | ARI ↑ | ACC ↑ | NMI ↑ | ARI ↑ | ACC ↑ | NMI ↑ | ARI ↑ | ACC ↑ | NMI ↑ | ARI ↑ |
| FMNIST | COMPLETER (Lin et al., 2021) | CVPR21 | 87.71 | 69.82 | 71.11 | 84.73 | 71.70 | 75.31 | 84.93 | 71.55 | 77.29 | 80.62 | 69.17 | 77.88 |
| | DSIMVC (Tang & Liu, 2022) | JMLR22 | 88.07 | 70.82 | 72.13 | 90.46 | 74.41 | 77.69 | 93.52 | 76.16 | 78.71 | 90.02 | 73.97 | 78.11 |
| | APADC (Xu et al., 2023) | TIP23 | 89.21 | 84.38 | 83.85 | 88.24 | 77.65 | 70.06 | 85.97 | 75.31 | 75.94 | 84.64 | 75.98 | 77.75 |
| | CPSPAN (Jin et al., 2023) | CVPR23 | 91.68 | 81.25 | 80.78 | 91.61 | 80.19 | 81.59 | 81.74 | 72.18 | 78.07 | 78.58 | 63.55 | 75.41 |
| | ProImp (Li et al., 2023) | IJCAI23 | 91.21 | 80.29 | 79.57 | 94.32 | 76.10 | 76.89 | 88.93 | 74.71 | 77.11 | 83.99 | 67.20 | 75.18 |
| | DIVIDE (Lu et al., 2024) | AAAI24 | 94.89 | 80.34 | 82.61 | 92.61 | 70.04 | 73.50 | 90.27 | 69.34 | 68.96 | 84.69 | 60.79 | 66.44 |
| | GIMVC (Bai et al., 2024) | CVDL24 | 93.65 | 79.98 | 81.06 | 92.85 | 74.32 | 74.45 | 91.16 | 72.64 | 70.66 | 82.87 | 63.47 | 69.01 |
| | DCG (Zhang et al., 2025a) | AAAI25 | 97.00 | 82.05 | 83.36 | 94.76 | 78.35 | 80.11 | 92.88 | 76.50 | 77.82 | 93.21 | 75.00 | 78.57 |
| | PMIMC (Yuan et al., 2025a) | TIP25 | 95.16 | 80.28 | 82.76 | 95.25 | 81.00 | 81.46 | 94.67 | 77.54 | 79.49 | 95.53 | 77.09 | 80.27 |
| | ROLL (Sun et al., 2025) | CVPR25 | 96.19 | 81.81 | 83.59 | 94.80 | 78.92 | 80.45 | 93.70 | 76.32 | 78.18 | 94.38 | 76.38 | 79.76 |
| | **MAGIC (Ours)** | Ours | **97.48** | **82.81** | **84.38** | **97.00** | **81.99** | **83.46** | **96.06** | **79.51** | **81.12** | **96.14** | **78.85** | **81.28** |
| Fashion | COMPLETER (Lin et al., 2021) | CVPR21 | 86.12 | 82.43 | 80.86 | 82.50 | 80.83 | 80.39 | 80.25 | 73.98 | 75.19 | 73.75 | 63.25 | 62.74 |
| | DSIMVC (Tang & Liu, 2022) | JMLR22 | 91.94 | 88.35 | 84.93 | 83.05 | 82.82 | 76.30 | 75.72 | 74.17 | 66.54 | 77.89 | 66.05 | 65.68 |
| | APADC (Xu et al., 2023) | TIP23 | 92.44 | 91.96 | 90.00 | 93.53 | 91.43 | 91.21 | 88.74 | 82.71 | 86.23 | 75.75 | 67.75 | 68.50 |
| | CPSPAN (Jin et al., 2023) | CVPR23 | 95.72 | 93.84 | 94.53 | 95.71 | 93.05 | 93.47 | 91.60 | 83.84 | 86.14 | 85.43 | 74.98 | 73.58 |
| | ProImp (Li et al., 2023) | IJCAI23 | 94.35 | 88.13 | 85.85 | 91.35 | 89.83 | 89.91 | 86.80 | 83.57 | 84.29 | 83.17 | 73.47 | 72.45 |
| | DIVIDE (Lu et al., 2024) | AAAI24 | 96.81 | 94.18 | 94.44 | 89.95 | 86.99 | 88.81 | 89.15 | 84.18 | 83.55 | 83.09 | 75.96 | 70.46 |
| | GIMVC (Bai et al., 2024) | CVDL24 | 97.54 | 95.04 | 95.69 | 93.75 | 90.85 | 91.48 | 91.24 | 86.96 | 85.24 | 82.16 | 73.21 | 71.07 |
| | DCG (Zhang et al., 2025a) | AAAI25 | 97.90 | 95.32 | 96.02 | 92.98 | 92.26 | 93.33 | 93.04 | 87.18 | 83.65 | 81.83 | 76.86 | 74.32 |
| | PMIMC (Yuan et al., 2025a) | TIP25 | 97.51 | 95.07 | 95.89 | 96.05 | 92.85 | 93.56 | 93.83 | 88.56 | 87.87 | 80.49 | 71.05 | 68.76 |
| | ROLL (Sun et al., 2025) | CVPR25 | 98.71 | 96.62 | 96.73 | 96.64 | 93.12 | 94.03 | 92.64 | 86.78 | 86.18 | 87.03 | 78.12 | 75.33 |
| | **MAGIC (Ours)** | Ours | **99.07** | **97.65** | **97.96** | **97.82** | **94.97** | **95.30** | **94.58** | **89.07** | **88.73** | **87.59** | **78.49** | **75.85** |

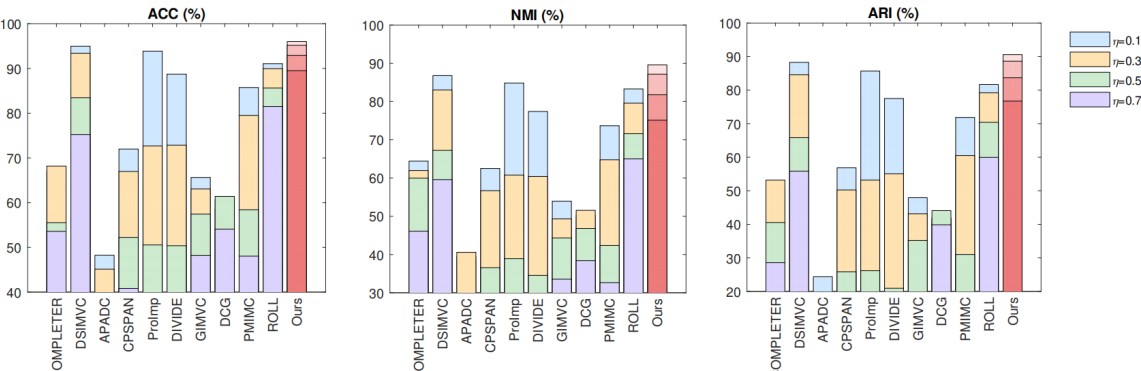

*Figure 3.* Comparison of clustering performance (ACC, NMI, and ARI) of different incomplete multi-view clustering methods under varying missing ratios $\eta \in \{0.1, 0.3, 0.5, 0.7\}$. Each bar reports the average performance over four benchmark datasets at the same missing ratio.

For fairness, each dataset–missing-rate pair uses one fixed missing mask shared by all methods, and every sample retains at least one observed view. All methods are run three times with different random seeds. ACC is computed after Hungarian label alignment, while NMI and ARI are computed using the standard scikit-learn implementations. The same metric routine and evaluation code path are used for all methods. Default hyperparameters, mask generation details, and implementation settings are provided in Appendix C.

### 4.3. Experimental Results and Analysis

Tables 1 and 2 report the clustering results on four IMVC benchmarks under missing ratios $\eta \in \{0.1, 0.3, 0.5, 0.7\}$. MAGIC obtains the highest ACC, NMI, and ARI in all reported settings. The advantage is more visible at larger missing ratios, where unreliable cross-view alignment and error propagation usually become more severe.

At $\eta = 0.7$, MAGIC improves the strongest baseline on BDGP from 89.64 to 90.79 in ACC and from 77.65 to 78.64 in ARI. On MNIST-USPS, MAGIC reaches 87.88 ACC, 76.36 NMI, and 75.59 ARI, compared with 86.80 ACC, 75.20 NMI, and 74.12 ARI from the strongest competitor. The same trend appears on larger datasets. On FMNIST, MAGIC obtains 96.14 ACC and 81.28 ARI, higher than the best baseline with 95.53 ACC and 80.27 ARI. On Fashion, MAGIC remains the top method across all missing ratios and improves the strongest baseline at $\eta = 0.7$ from 87.03 ACC, 78.12 NMI, and 75.33 ARI to 87.59 ACC, 78.49 NMI, and 75.85 ARI.

These results suggest that MAGIC loses performance more slowly as $\eta$ increases, while keeping stable clustering quality across datasets with different scales and view configurations. Fig. 3 gives the averaged ACC, NMI, and ARI across benchmarks at each missing ratio. The t-SNE visual-

*Table 2.* Clustering performance (%) under different missing ratios $\eta$ (continued).

| Dataset | Method | References | $\eta = 0.1$ | | | $\eta = 0.3$ | | | $\eta = 0.5$ | | | $\eta = 0.7$ | | |
|---|---|---|---|---|---|---|---|---|---|---|---|---|---|---|
| | | | ACC ↑ | NMI ↑ | ARI ↑ | ACC ↑ | NMI ↑ | ARI ↑ | ACC ↑ | NMI ↑ | ARI ↑ | ACC ↑ | NMI ↑ | ARI ↑ |
| BDGP | COMPLETER (Lin et al., 2021) | CVPR21 | 89.16 | 88.42 | 86.23 | 85.60 | 80.34 | 86.02 | 89.12 | 78.68 | 79.06 | 80.48 | 66.15 | 66.89 |
| | DSIMVC (Tang & Liu, 2022) | JMLR22 | 96.16 | 93.68 | 95.46 | 95.28 | 85.75 | 88.51 | 78.96 | 69.53 | 72.61 | 80.68 | 67.22 | 67.51 |
| | APADC (Xu et al., 2023) | TIP23 | 89.84 | 85.91 | 82.88 | 85.16 | 82.87 | 73.93 | 80.88 | 75.13 | 78.40 | 85.96 | 63.31 | 61.01 |
| | CPSPAN (Jin et al., 2023) | CVPR23 | 88.00 | 87.63 | 81.36 | 82.24 | 80.57 | 72.78 | 83.92 | 71.61 | 74.42 | 81.56 | 68.67 | 63.38 |
| | ProImp (Li et al., 2023) | IJCAI23 | 95.00 | 87.32 | 88.11 | 86.84 | 84.94 | 89.76 | 81.76 | 75.72 | 77.87 | 83.96 | 63.93 | 65.07 |
| | DIVIDE (Lu et al., 2024) | AAAI24 | 90.53 | 88.36 | 85.65 | 85.93 | 88.59 | 89.35 | 85.45 | 75.84 | 76.71 | 85.22 | 65.63 | 68.80 |
| | GIMVC (Bai et al., 2024) | CVDL24 | 86.12 | 87.42 | 81.86 | 94.24 | 87.59 | 87.32 | 89.40 | 80.95 | 81.00 | 80.08 | 73.79 | 73.71 |
| | DCG (Zhang et al., 2025a) | AAAI25 | 97.84 | 94.63 | 95.71 | 94.28 | 87.78 | 88.22 | 87.20 | 78.41 | 81.42 | 88.76 | 70.17 | 65.12 |
| | PMIMC (Yuan et al., 2025a) | TIP25 | 98.44 | 95.56 | 96.65 | 95.56 | 88.79 | 90.38 | 93.36 | 82.30 | 85.18 | 89.64 | 74.76 | 77.65 |
| | ROLL (Sun et al., 2025) | CVPR25 | 97.96 | 94.42 | 95.00 | 95.60 | 88.21 | 90.73 | 92.60 | 81.28 | 83.52 | 88.36 | 74.89 | 77.31 |
| | **MAGIC (Ours)** | Ours | **98.88** | **96.30** | **97.20** | **97.04** | **90.51** | **92.74** | **94.33** | **83.62** | **86.54** | **90.79** | **75.32** | **78.64** |
| MNIST-USPS | COMPLETER (Lin et al., 2021) | CVPR21 | 97.00 | 94.44 | 93.21 | 97.40 | 94.01 | 94.49 | 76.80 | 81.67 | 72.36 | 76.62 | 64.88 | 61.17 |
| | DSIMVC (Tang & Liu, 2022) | JMLR22 | 97.90 | 94.84 | 95.41 | 90.68 | 87.94 | 88.74 | 92.02 | 82.83 | 86.10 | 80.90 | 67.08 | 67.49 |
| | APADC (Xu et al., 2023) | TIP23 | 93.22 | 94.53 | 91.68 | 92.48 | 91.97 | 90.80 | 88.68 | 84.13 | 82.34 | 79.16 | 65.80 | 65.51 |
| | CPSPAN (Jin et al., 2023) | CVPR23 | 94.26 | 93.27 | 93.07 | 93.02 | 93.89 | 91.06 | 89.06 | 85.64 | 86.23 | 79.52 | 66.04 | 65.25 |
| | ProImp (Li et al., 2023) | IJCAI23 | 96.68 | 92.17 | 92.71 | 92.98 | 91.28 | 90.93 | 89.44 | 83.96 | 81.59 | 83.96 | 74.39 | 73.70 |
| | DIVIDE (Lu et al., 2024) | AAAI24 | 96.98 | 92.50 | 93.42 | 92.10 | 90.08 | 91.10 | 90.00 | 83.05 | 84.10 | 81.98 | 71.24 | 71.19 |
| | GIMVC (Bai et al., 2024) | CVDL24 | 96.15 | 93.38 | 94.36 | 92.12 | 92.69 | 91.57 | 87.74 | 80.12 | 81.41 | 83.32 | 72.89 | 71.98 |
| | DCG (Zhang et al., 2025a) | AAAI25 | 96.64 | 94.16 | 94.60 | 93.08 | 90.32 | 91.95 | 89.86 | 82.65 | 85.34 | 76.04 | 74.23 | 71.54 |
| | PMIMC (Yuan et al., 2025a) | TIP25 | 97.58 | 94.95 | 95.09 | 96.48 | 93.78 | 94.50 | 93.12 | 89.60 | 87.20 | 86.80 | 75.20 | 74.12 |
| | ROLL (Sun et al., 2025) | CVPR25 | 97.35 | 94.85 | 95.06 | 95.64 | 92.55 | 93.52 | 92.36 | 85.35 | 86.70 | 85.93 | 74.30 | 73.53 |
| | **MAGIC (Ours)** | Ours | **98.02** | **95.03** | **95.67** | **97.80** | **94.33** | **95.19** | **94.72** | **87.34** | **88.68** | **87.88** | **76.36** | **75.59** |

BDGP (PMIMC)  BDGP (Ours)  MNIST-USPS (PMIMC)  MNIST-USPS (Ours)  FMNIST (PMIMC)  FMNIST (Ours)

BDGP (PMIMC)  BDGP (Ours)  MNIST-USPS (PMIMC)  MNIST-USPS (Ours)  FMNIST (PMIMC)  FMNIST (Ours)

*Figure 4.* t-SNE visualizations of clustering results under different missing rates ($\eta \in \{0.3, 0.7\}$) for the datasets BDGP, MNIST-USPS, and FMNIST. Each pair of images (left: PMIMC, right: Ours) represents the clustering results of the same dataset under the corresponding missing rate.

*Table 3.* Ablation of loss terms (%) on two datasets.

| Loss terms | | | BDGP ($\eta = 0.1$) | | FMNIST ($\eta = 0.5$) | |
|---|---|---|---|---|---|---|
| $L_{\text{rec}}$ | $L_{\text{con}}$ | $L_{\text{al}}$ | ACC ↑ | NMI ↑ | ACC ↑ | NMI ↑ |
| | | | 61.04 | 60.46 | 68.68 | 62.09 |
| ✓ | | | 65.15 | 63.52 | 65.58 | 50.66 |
| | ✓ | | 70.37 | 70.18 | 71.82 | 52.80 |
| | | ✓ | 85.75 | 83.36 | 75.60 | 57.44 |
| ✓ | ✓ | | 89.75 | 87.27 | 83.90 | 69.59 |
| ✓ | | ✓ | 90.00 | 88.38 | 88.55 | 71.62 |
| | ✓ | ✓ | 95.29 | 92.13 | 93.35 | 77.14 |
| ✓ | ✓ | ✓ | **98.88** | **96.30** | **96.06** | **79.51** |

## 4.4. Ablation Analysis

Table 3 ablates the three objectives in MAGIC: masked reconstruction $L_{\text{rec}}$, multi-path consensus $L_{\text{con}}$, and cross-view alignment $L_{\text{al}}$, on BDGP ($\eta = 0.1$) and FMNIST ($\eta = 0.5$). Starting from the variant with no objective enabled, adding a single loss improves performance in most cases, but the effect differs across datasets.

On BDGP, $L_{\text{al}}$ alone gives the largest single-term gain, increasing ACC from 61.04 to 85.75 and NMI from 60.46 to 83.36. This suggests that cross-view agreement is already useful under mild missingness. By contrast, $L_{\text{rec}}$ and $L_{\text{con}}$ alone give smaller gains. On FMNIST, $L_{\text{rec}}$ alone brings limited benefit and even underperforms the plain variant in ACC, while combinations of different objectives give

izations in Fig. 4 show more compact and better-separated clusters for MAGIC. The visualization protocol is detailed in Appendix C.

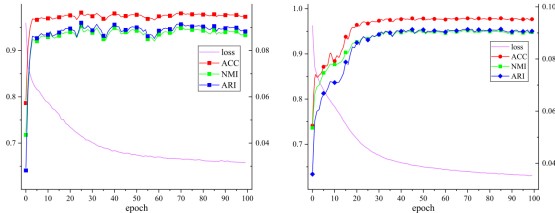

*Figure 5.* Convergence curves of ACC, NMI, and ARI on MNIST-USPS and Fashion under $\eta = 0.3$.

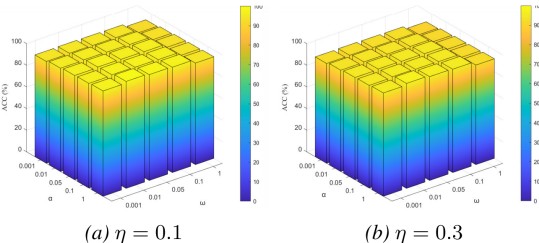

*(a) $\eta = 0.1$*        *(b) $\eta = 0.3$*

*Figure 6.* Sensitivity of MAGIC to the loss weights $\alpha$ and $\omega$ on the BDGP dataset.

clearer improvements. The strongest two-term variant is $L_{con} + L_{al}$, which reaches 93.35 ACC and 77.14 NMI on FMNIST. Using all three objectives gives the best results on both datasets, with 98.88 ACC and 96.30 NMI on BDGP, and 96.06 ACC and 79.51 NMI on FMNIST. This pattern indicates that reconstruction, consensus calibration, and alignment provide complementary signals.

### 4.5. Convergence Analysis

To examine optimization stability, we report representative training curves on MNIST-USPS ($\eta = 0.3$) and Fashion ($\eta = 0.3$) in Fig. 5. On both datasets, the objective decreases rapidly in the early epochs and then approaches a flat region without abrupt oscillations. ACC, NMI, and ARI rise quickly at the beginning, which corresponds to the formation of coarse cluster structure, and then plateau with mild fluctuations.

The trends of the objective and the clustering metrics are also consistent with the training design of MAGIC. In the early stage, multi-path consensus provides strong alignment signals before semantic imputation is fully activated. After the posteriors become more stable, confidence-gated imputation further refines missing-view semantics without causing evident metric degradation. These curves show that MAGIC can be trained stably, and that multi-path agreement does not introduce noticeable optimization instability. Additional convergence results are provided in the appendix.

### 4.6. Parameter Sensitivity Analysis

We analyze the sensitivity of MAGIC to the loss weights $\alpha$ and $\omega$ in Eq. (15) on BDGP under mild and moderate missingness ($\eta \in \{0.1, 0.3\}$). The results are shown in Fig. 6. We sweep $(\alpha, \omega)$ over $\{0.001, 0.01, 0.05, 0.1, 1\}$ while keeping all other hyperparameters fixed. The ACC surfaces are smooth and exhibit a broad high-performance region around the default setting, indicating low sensitivity to moderate changes in either coefficient.

For both $\eta = 0.1$ and $\eta = 0.3$, the best region lies near the default configuration, and nearby choices lead to only minor variations. This suggests that MAGIC does not rely on a narrow set of hyperparameters to obtain strong performance. The consensus term weighted by $\alpha$ improves posterior stability, while the alignment regularizer weighted by $\omega$ encourages compatible view-wise semantics. Performance drops mainly at extreme settings where one term dominates, which can over-emphasize agreement or over-regularize alignment at the expense of separability. Overall, MAGIC remains stable over a wide range of $(\alpha, \omega)$. Additional guidance for label-free parameter selection is given in Appendix C.

## 5. Conclusion

We studied incomplete multi-view clustering under high missingness, where sparse co-observations and view imbalance make alignment fragile and early imputation may amplify errors. MAGIC first stabilizes representation learning through multi-path contrastive consensus, and then performs confidence-gated semantic completion to reduce error propagation. This design separates semantic calibration from missing-view completion, helping prevent unreliable early assignments from dominating later training. Across four benchmarks, MAGIC gives consistently higher clustering scores than recent baselines. The ablation results further show that reconstruction, consensus learning, and alignment provide complementary signals. The convergence and sensitivity analyses indicate that the proposed objective can be optimized stably and is not overly sensitive to moderate changes in key loss weights. Future work will extend MAGIC to partially paired settings and stronger uncertainty modeling under distribution shift.

## Impact Statement

This paper studies incomplete multi-view clustering under partial and imbalanced observations. The method may be useful for multimodal data integration and exploratory analysis when some views are unavailable. As with other clustering methods, applying it to personal or sensitive data may raise privacy and profiling risks. We recommend privacy protection, fairness checks, and careful validation before deployment. We do not identify additional societal risks beyond these general concerns for unsupervised learning.

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

## A. Additional Theoretical Results

**Proposition A.1** (InfoNCE lower bound on mutual information). *Let $(U, V) \sim p(u, v)$ with marginals $p(u)$ and $p(v)$, and let $M \geq 2$ be the number of candidates, including one positive sample and $M - 1$ negative samples. Sample $(u, v_1) \sim p(u, v)$ and draw $\{v_j\}_{j=2}^M$ independently from $p(v)$. For any measurable score function $s(u, v) > 0$, define the population InfoNCE objective*

$$\mathcal{L}_{\text{NCE}} = -\mathbb{E}\left[\log \frac{s(u, v_1)}{\sum_{j=1}^M s(u, v_j)}\right]. \tag{17}$$

*It is shown in (Oord et al., 2018) that*

$$I(U; V) \geq \log M - \mathcal{L}_{\text{NCE}}. \tag{18}$$

**Proposition A.2** (Entropic OT: existence and Sinkhorn scaling). *Let $\text{Cost} \in \mathbb{R}^{K \times K}$ and let $a, b \in \mathbb{R}_+^K$ satisfy $a^\top \mathbf{1} = b^\top \mathbf{1} = 1$ and $a_k > 0, b_k > 0$ for all $k$. For $\varepsilon > 0$, the problem in Eq. (10) admits a unique minimizer $\Pi^\star$ and has the Sinkhorn scaling form established in (Cuturi, 2013):*

$$\Pi^\star = \text{diag}(u)\, K\, \text{diag}(v), \qquad K_{kk'} = \exp(-\text{Cost}_{kk'}/\varepsilon), \tag{19}$$

*for some $u, v \in \mathbb{R}_+^K$ such that $\Pi^\star \mathbf{1} = a$ and $(\Pi^\star)^\top \mathbf{1} = b$.*

**Proof of Theorem 3.1.** Since $T$ is row-stochastic, it defines a Markov kernel that maps a distribution $p$ to $pT$. The data-processing inequality for KL divergence states that applying the same Markov kernel to two distributions cannot increase their KL divergence. Therefore,

$$\text{KL}(pT \| qT) \leq \text{KL}(p \| q).$$

Applying the same argument after exchanging $p$ and $q$ gives

$$\text{KL}(qT \| pT) \leq \text{KL}(q \| p).$$

Adding the two inequalities yields

$$\text{SKL}(pT, qT) \leq \text{SKL}(p, q).$$

This completes the proof.

## B. Additional Details for Confidence-Gated Imputation

**Sharpening.** Given $Q^v = [q_{nk}^v] \in \mathbb{R}^{N \times K}$, define the DEC-style sharpening

$$\tilde{q}_{nk}^v = \frac{(q_{nk}^v)^2 \big/ \sum_{i=1}^N q_{ik}^v}{\sum_{t=1}^K (q_{nt}^v)^2 \big/ \sum_{i=1}^N q_{it}^v}, \qquad \tilde{q}_{n:}^v \in \Delta^{K-1}.$$

**View similarity.** We compute $s_{vu} \in [0, 1]$ in label space using only co-observed instances of views $v$ and $u$. In our implementation, the similarity is obtained by averaging the cosine similarity between the sharpened posteriors of the co-observed samples and then clipping the result into $[0, 1]$ when needed. No missing sample is used to estimate the view-pair similarity.

**Optional kNN refinement.** For latent-space refinement, let $k_{\text{nn}} \in \mathbb{N}$ be the neighbor count and let $d_{\text{nn}}(\cdot, \cdot)$ be a fixed distance on $\mathbb{R}^d$, such as cosine distance. For cluster $c$ in view $v$, define the visible set

$$\mathcal{V}_c^v = \{r \mid m_r^v = 1 \text{ and } c^\star(r) = c\}.$$

If $|\mathcal{V}_{c^\star(n)}^v| \geq k_{\text{nn}}$, we may replace Eq. (14) by averaging the $k_{\text{nn}}$ nearest neighbors of $n$ in $\mathcal{V}_{c^\star(n)}^v$ under $d_{\text{nn}}$.

## C. Implementation and Evaluation Details

**Code Availability.** The source code is available at https://github.com/sakulalu/MAGIC.

**Alignment implementation.** The alignment loss in Eq. (16) is computed only for view pairs with non-empty co-observed sets. If $\mathcal{O}_{vu}$ is empty, that view pair is skipped. This avoids introducing gradients from raw missing entries and keeps the alignment signal restricted to reliable co-observed evidence.

**Missing mask generation.** For each dataset and missing ratio, the missing mask is generated once and reused for all compared methods. We use a constrained random protocol rather than independent Bernoulli sampling for each view. Each sample is required to retain at least one observed view. For partially observed samples, one preserved view is first assigned by one-hot sampling, and additional observed entries are randomly added so that the final observed-view ratio matches the target missing ratio as closely as possible. The sample order is then randomly permuted to form the final mask matrix.

**Evaluation protocol.** All methods use the same evaluation pipeline. ACC is computed after optimal label-permutation alignment using the Hungarian assignment algorithm. NMI and ARI are computed using the standard scikit-learn implementations.

**Default settings.** The default implementation settings used in the main experiments are summarized in Table 4. The masked-fusion branch applies random masking at the feature-dimension level. Prototypes are recomputed online from the current mini-batch and visible samples, without a memory bank or momentum update.

*Table 4.* Default implementation settings used in the main experiments.

| Item | Default setting |
| --- | --- |
| Reconstruction pretraining | 100 epochs |
| Contrastive fine-tuning | 100 epochs |
| OT semantic imputation start | Epoch 30 |
| kNN refinement start | Epoch 50 |
| Gating threshold $\tau_{\mathrm{src}}$ | linearly from 0.65 to 0.55 |
| Aggregation temperature $\tau_{\mathrm{agg}}$ | 0.8 |
| Path weights $(\pi_f, \pi_p, \pi_m)$ | $(1/3, 1/6, 1/2)$ |
| Label contrastive temperature | 0.7, and 0.5 on Fashion |
| Masked-fusion granularity | feature dimension |
| Masked-fusion ratio | 0.7 |
| Masked-fusion samples | 6 |

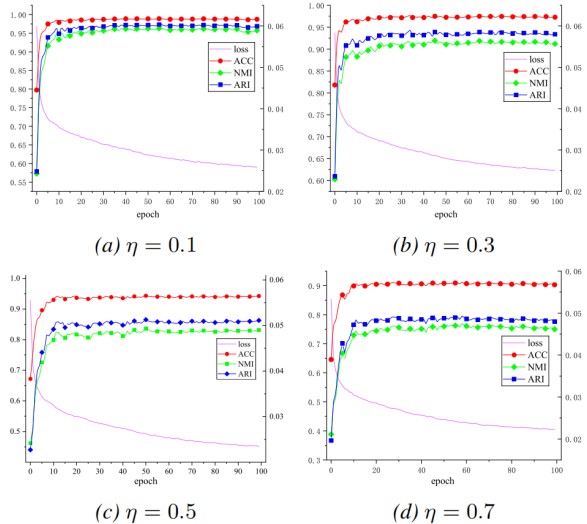

*(a)* $\eta = 0.1$        *(b)* $\eta = 0.3$

*(c)* $\eta = 0.5$        *(d)* $\eta = 0.7$

*Figure 7.* Additional convergence curves of MAGIC on BDGP under different missing ratios. Each subplot reports the training loss and clustering metrics, including ACC, NMI, and ARI.

**Visualization protocol.** All t-SNE visualizations are generated from the fused representation rather than a single-view embedding. For datasets with more than 5000 samples, we first reduce the representation to 50 dimensions by PCA and then apply t-SNE. Otherwise, t-SNE is applied directly to the fused representation. The same visualization script and hyperparameters are used across methods and missing-rate settings.

**Label-free parameter guidance.** For label-free use, we recommend starting from the default weights and the conservative gating schedule. If posterior entropy remains high or cross-path agreement is unstable, the semantic-transfer gate should remain strict. If posteriors have stabilized but imputation remains weak, the gate can be relaxed later in training. In general, the consensus-learning term should remain the dominant component, while the alignment regularizer should be kept in a moderate range to avoid over-regularization.

# D. Additional Experimental Results

### D.1. Additional Convergence Results

We report additional convergence curves on BDGP under four missing ratios. Each subplot shows the training loss together with ACC, NMI, and ARI. These curves supplement the convergence analysis in the main paper and show that MAGIC maintains stable training behavior as the missing ratio increases.

### D.2. Additional Visualization Results

We provide additional t-SNE visualizations of MAGIC on Fashion under different missing ratios. These figures follow the visualization protocol described in Appendix C and complement the qualitative results in the main paper.

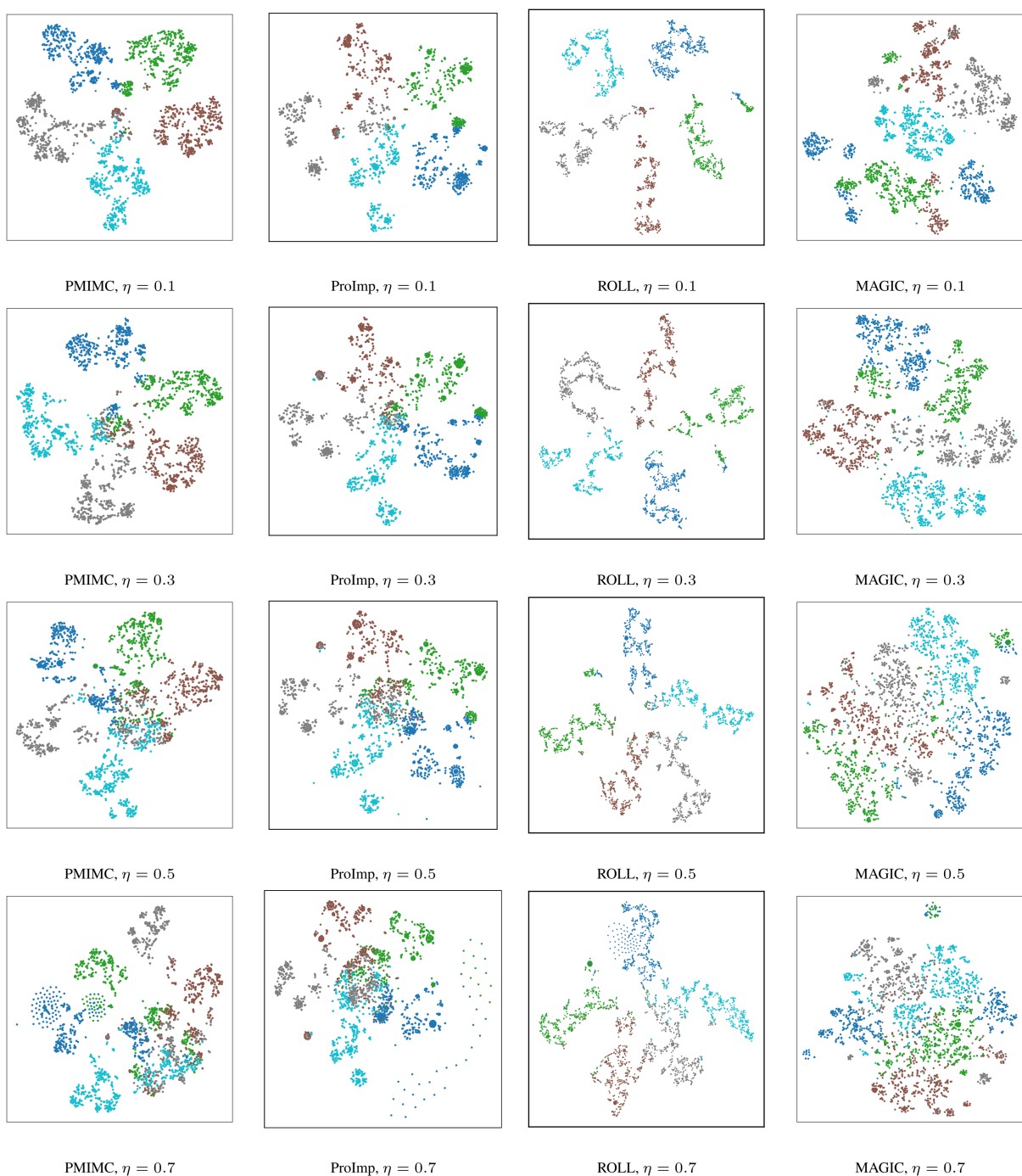

*Figure 8.* Additional t-SNE visualizations on BDGP under different missing ratios. Rows correspond to $\eta = 0.1$, $\eta = 0.3$, $\eta = 0.5$, and $\eta = 0.7$. Columns correspond to PMIMC, ProImp, ROLL, and MAGIC.

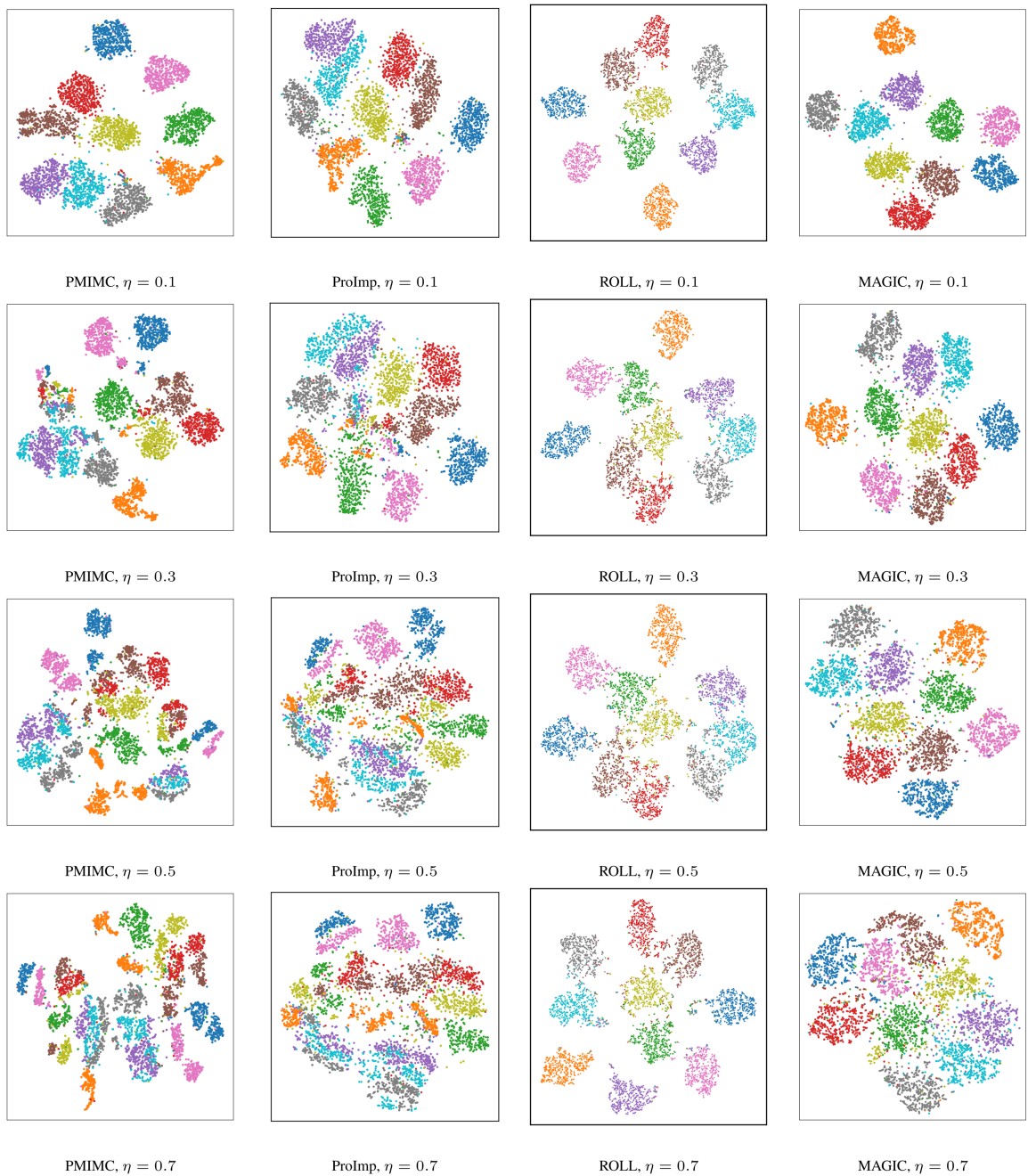

*Figure 9.* Additional t-SNE visualizations on MNIST-USPS under different missing ratios. Rows correspond to $\eta = 0.1$, $\eta = 0.3$, $\eta = 0.5$, and $\eta = 0.7$. Columns correspond to PMIMC, ProImp, ROLL, and MAGIC.

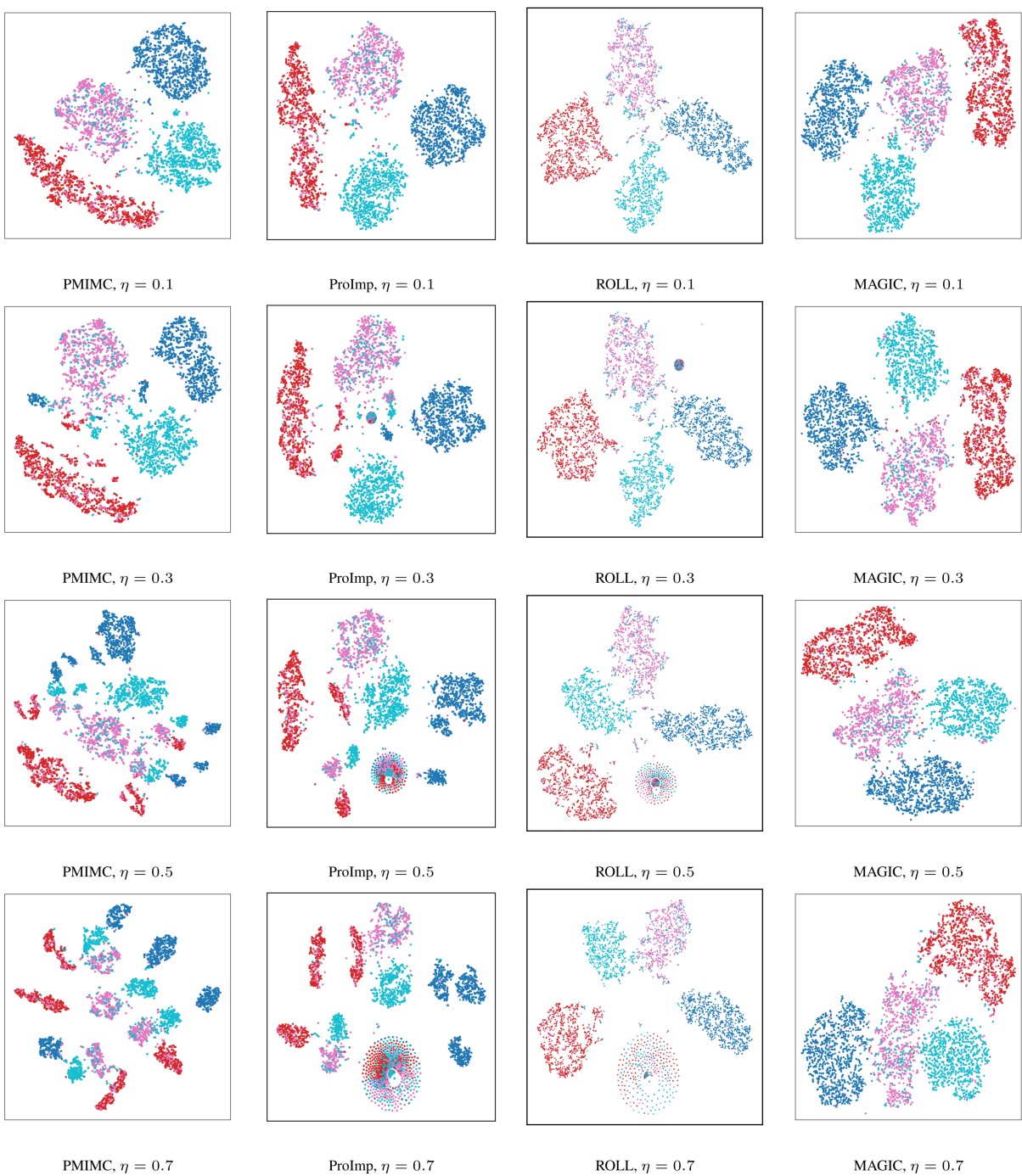

*Figure 10.* Additional t-SNE visualizations on FMNIST under different missing ratios. Rows correspond to $\eta = 0.1$, $\eta = 0.3$, $\eta = 0.5$, and $\eta = 0.7$. Columns correspond to PMIMC, ProImp, ROLL, and MAGIC.

