# OpenReview forum: "Masked Multi-path Contrast with Confidence-Gated Semantic Imputation for Incomplete Multi-view Clustering"
_ICML.cc/2026/Conference — ICML 2026 regular_

### Official Review · Reviewer_uYjd · 2026-02-25

**Soundness:** 4
**Presentation:** 3
**Significance:** 3
**Originality:** 3
**Overall Recommendation:** 5
**Confidence:** 4

**Summary:**

MAGIC improves robustness for incomplete multi-view clustering under high missingness and view-visibility imbalance via a two-stage design. Stage one learns fused, per-view, and masked-fusion paths and enforces cross-path consensus using contrastive objectives and prediction-consistency regularization, stabilizing posteriors when co-observed pairs are scarce. Stage two performs uncertainty-aware propagation and imputation, where normalized-entropy gating activates class-level semantic transfer only under sufficient confidence and prototype fallback constrains missing-view representations.

**Compliance With Llm Reviewing Policy:**

Affirmed.

**Final Justification:**

The authors did a comprehensive response and I'm maintaining my original score.

**Key Questions For Authors:**

1. The paper uses view-wise posteriors, an aggregated consensus posterior, and a post-transfer posterior (e.g., $\hat{q}$). Please provide a brief mapping from each symbol to its meaning and indicate at which stage each quantity is computed and used.

2. When aggregating posteriors across the three paths, how are the path weights and any temperature parameters set by default? Are they fixed across datasets or tuned per dataset? Please report the concrete default values used in the main results.

3. Cross-view alignment is described as an important term. Is this alignment computed strictly on co-observed sample–view pairs, or can any gradients propagate through missing entries? Clarifying the effective sample set would improve reproducibility.

4. In label-free deployments, how should $\alpha$, $\omega$, and the gating threshold be selected in practice? For example, recommended default ranges or a simple heuristic procedure (e.g., start from defaults and monitor posterior entropy and cross-path consistency) would be helpful.

**Limitations:**

Yes

**Strengths And Weaknesses:**

(1) The paper identifies two key IMVC tensions, scarce co observed pairs that weaken alignment and pseudo label diffusion that causes cascading errors, and addresses them via a stabilize then controlled propagate design supported by ablations and diagnostics.

(2) The method is easy to follow. Multi path learning mitigates semantic instability and dominant view pull, while gated imputation curbs propagation errors. A structured comparison to adjacent work would better highlight the key differentiators.

(3) Experiments cover low to extreme missing rates. Performance degrades mildly as missingness increases, and ablations, convergence curves, and sensitivity plots support stable optimization and practical robustness under imbalanced view visibility.

(4) While components are related to known ideas, novelty comes from integrating them into a risk aware pipeline targeting distinct failure modes across stages. A concise comparative summary would further clarify what is new and when each module matters.

(5) Some implementation details should be fixed in the main text, including the alignment regularizer form, aggregation weights or temperature, and masked fusion mask settings. Guidance for choosing alpha, omega, and gating thresholds without labels would improve usability.

---

> ### Author Rebuttal · Authors · 2026-03-30
>
> We thank Reviewer `uYjd` for the careful review, positive assessment, and constructive questions. We especially appreciate the reviewer’s recognition that **MAGIC addresses two key failure modes under high missingness: fragile alignment under scarce co-observed pairs and error amplification from premature semantic propagation, through a two-stage design that first stabilizes posteriors and then performs controlled semantic transfer and imputation**. We also thank the reviewer for highlighting the complementary roles of multi-path learning, posterior aggregation, confidence gating, and prototype fallback, as well as the strong support from the ablations, convergence curves, and sensitivity analyses. To address the reviewer’s questions and the points where additional clarification would strengthen reproducibility and usability, we respond directly below.
>
> **Q1.** The paper uses view-wise posteriors, an aggregated consensus posterior, and a post-transfer posterior. Please provide a brief mapping from each symbol to its meaning and indicate at which stage each quantity is computed and used.
>
> **Respond:** The **path-wise posteriors** are the stage-1 outputs before aggregation: fused-path, per-view, and masked-fusion posteriors. The **view-wise consensus posterior** aggregates them for each view and is the **calibrated stage-1 output**. In stage 2, it is sharpened to compute **class masses**, **class prototypes**, and **view similarity**. The **post-transfer posterior** is the target-view posterior after **confidence-gated class-level transfer** from the selected source view; it is then used for imputation and, when needed, prototype or kNN refinement. We now state this mapping explicitly in the method section.
>
> **Q2.** When aggregating posteriors across the three paths, how are the path weights and any temperature parameters set by default? Are they fixed across datasets or tuned per dataset? Please report the concrete default values used in the main results.
>
> **Respond:** The mixing coefficients in **Eq. (5)** and the aggregation temperature are **fixed** in the main results. The effective normalized coefficients for the **fused**, **per-view**, and **masked-fusion** paths are **$1/3$**, **$1/6$**, and **$1/2$**, respectively, and the aggregation temperature is **$0.8$**. These coefficients are **implicitly normalized** by dividing by their sum in the barycentric log-opinion aggregation and are **fixed across datasets**. The label-level contrastive temperature is **$0.7$ by default** and **$0.5$ on Fashion**. We now include these defaults in the experimental setup.
>
> **Q3.** Cross-view alignment is described as an important term. Is this alignment computed strictly on co-observed sample–view pairs, or can any gradients propagate through missing entries? Clarifying the effective sample set would improve reproducibility.
>
> **Respond:** The effective sample set is **term-specific**. Reconstruction is applied **only on observed entries**, so missing entries contribute **no reconstruction gradients**. For **cross-view alignment**, the pairwise view-similarity coefficient is computed **strictly on the co-observed subset** of that view pair, and this pair-specific coefficient weights the alignment term between the corresponding view-wise posteriors. Thus, the effective alignment signal comes from **co-observed samples**, not missing entries; there is **no single global co-observed index** shared by all losses. We now clarify this directly in the method description.
>
> **Q4.** In label-free deployments, how should $\alpha$, $\omega$, and the gating threshold be selected in practice? For example, recommended default ranges or a simple heuristic procedure would be helpful.
>
> **Respond:** For label-free use, we recommend starting from the main settings: **$\alpha = 0.1$**, **$\omega = 0.01$**, and a conservative **$\tau_{\mathrm{src}}$** schedule from **$0.65$ to $0.55$**. In practice, adjust only based on **posterior entropy** and **cross-path consistency**: if entropy remains high or agreement is unstable, keep the gate more conservative; if posteriors have stabilized but imputation remains too weak, relax the gate later. More generally, the **main consensus-learning component** should remain dominant, while auxiliary alignment and regularization terms use smaller or comparable weights. We now add this practical guidance to the experimental-details discussion.
>
> We thank Reviewer `uYjd` again for recognizing both the technical contribution and the empirical strength of the paper. We hope the revised clarifications make the contribution, symbol mapping, default settings, and reproducibility of MAGIC much clearer.

---

> > ### Author Rebuttal · Reviewer_uYjd · 2026-04-02
> >
> > The author did a comprehensive response and I'm maintaining my original score.

---

### Official Review · Reviewer_aqnR · 2026-02-25

**Soundness:** 4
**Presentation:** 3
**Significance:** 3
**Originality:** 4
**Overall Recommendation:** 6
**Confidence:** 4

**Summary:**

This paper targets robustness in incomplete multi-view clustering under high missing rates and proposes MAGIC, a framework that first estimates a relatively reliable clustering semantic distribution and then uses it for controlled semantic propagation and missing-view imputation. Specifically, a) during representation learning, MAGIC constructs multiple correlated learning paths and reduces cross-path disagreement via contrastive consensus constraints together with prediction-consistency regularization; b) during imputation, it applies an entropy-based confidence gate to trigger class-level semantic transfer only when the confidence criterion is met, and falls back to class prototypes to represent missing views when evidence is insufficient; c) empirically, the evaluation covers multiple datasets and missing-rate settings, and includes ablations, convergence curves, and sensitivity analyses that collectively support the method’s effectiveness and training stability.

**Compliance With Llm Reviewing Policy:**

Affirmed.

**Final Justification:**

The authors provided a detailed response to the main concerns, which has solved all my concerns. Therefore, I'm maintaining my original score and recommend a strong accept.

**Key Questions For Authors:**

1. For the contrastive consensus loss, prediction-consistency regularizer, and cross-view alignment regularizer, what are the exact effective sample sets used in practice? In particular, are these losses computed strictly on co-observed sample–view pairs, and is the co-observed index set shared across the fused/per-view/masked-fusion paths or defined separately per term/path?
2. In Eq. (13), how is the class prototype $C^v$ computed and updated in the implementation? For example, is it an epoch-wise mean over observed samples, an EMA-smoothed statistic, or a learnable parameter, and is it updated per batch or per epoch (with any normalization)?
3. At test time, when multiple views are observed for a sample, what is the default strategy used to output the final label in all reported results? Is it based on the fused posterior, a fixed view posterior, or an aggregation over the observed views’ posteriors?

**Limitations:**

Yes

**Strengths And Weaknesses:**

1. Under high missingness, a key failure mode is repeated propagation of uncertain early semantics that gradually biases learning. The paper addresses this by delaying imputation via a training schedule and using confidence gating to limit when/where semantics are propagated, improving robustness and reducing error cascades.
2. The confidence signal is interpretable and easy to reason about. Using normalized entropy to gate semantic transfer prevents forcing unreliable information into missing views. The prototype fallback further constrains the space of missing-view representations, which helps limit imputation drift and reduces overfitting risk when boundaries are ambiguous. Some symbol/set definitions could still be made more explicit to improve reproducibility.
3. The experimental evaluation is solid: results span a broad range of missing rates and include a focused summary for the hardest regime. Ablations support that objective components are complementary, while convergence curves and hyper-parameter sensitivity analyses indicate smooth optimization and reasonable robustness around default settings.
4. The mechanism-level story is consistent. Multi-path learning reduces dependence on scarce co-observed pairs, and aggregation with consistency constraints improves posterior reliability. Confidence gating regulates semantic transfer, and prototype fallback mitigates drift. It would be helpful to clarify whether each loss term is strictly restricted to co-observed pairs and whether any gradients propagate through missing entries, as this can materially affect stability under extreme missingness.
5. The contribution, novelty, and evidence support a strong accept, but the final draft should remove residual template text and fix minor caption typos such as “Fashion and Fashion.”

---

> ### Author Rebuttal · Authors · 2026-03-30
>
> We thank Reviewer `aqnR` very much for the **strong accept** recommendation and for clearly recognizing the core innovation of our paper. We especially appreciate the reviewer’s summary that **MAGIC first stabilizes semantic posteriors through multi-path consensus learning and then performs controlled semantic propagation and imputation through confidence gating and prototype fallback**. We also thank the reviewer for highlighting the consistency of the mechanism-level story and for recognizing that the experiments, ablations, convergence curves, and sensitivity analyses provide strong evidence for both effectiveness and training stability. To address the reviewer’s questions and the points where further clarification would improve reproducibility, we respond directly below.
>
> **Q1.** For the contrastive consensus loss, prediction-consistency regularizer, and cross-view alignment regularizer, what are the exact effective sample sets used in practice? In particular, are these losses computed strictly on co-observed sample–view pairs, and is the co-observed index set shared across the fused path, the per-view path, and the masked-fusion path, or defined separately for each term and path?
>
> **Respond:** The effective sample sets are **term-specific**, not governed by one shared global co-observed index. For the multi-path contrastive objective, the **fused–fused term** is computed on **all samples in the batch**; the **per-view–fused terms** are computed **only on samples where that specific view is visible**; and the **masked-fusion–fused terms** are also computed on **all samples**, after masked fusion is formed from the available views. The **prediction-consistency regularizer** is applied to paired posteriors from the two augmentations for the fused path, the masked-fusion path, and each per-view path. For **cross-view alignment**, the pairwise view-similarity coefficient is computed **strictly on the co-observed subset** of that view pair, and this pair-specific coefficient then weights the alignment term between the corresponding view-wise posteriors. Reconstruction is applied **only on observed entries**, so **missing entries do not contribute reconstruction gradients**.
>
> **Q2.** In Eq. (13), how is the class prototype computed and updated in the implementation? For example, is it an epoch-wise mean over observed samples, an EMA-smoothed statistic, or a learnable parameter, and is it updated per batch or per epoch, with any normalization?
>
> **Respond:** The class prototype is **not** a learnable parameter and is **not** EMA-smoothed. It is **recomputed online from the current mini-batch in each forward pass**. For each view, we take the current feature matrix and soft assignments, mask out missing samples using the visibility indicator, and compute each cluster prototype as the **soft-assignment-weighted mean of the currently observed samples** in that batch. The result is normalized by the corresponding class mass, with a small constant for numerical stability. Thus, the prototype is a **batch-wise weighted centroid over observed samples**, updated **every forward pass rather than every epoch**. In the revised manuscript, we now state this explicitly around Eq. (13), including that the prototype is **batch-wise**, **observation-masked**, **soft-assignment-weighted**, and **non-learnable**.
>
> **Q3.** At test time, when multiple views are observed for a sample, what is the default strategy used to output the final label in all reported results? Is it based on the fused posterior, a fixed view posterior, or an aggregation over the posteriors of the observed views?
>
> **Respond:** The final prediction is **not** taken from the raw fused posterior alone and is **not** taken from any fixed single-view posterior. Instead, the default inference rule is a **confidence-weighted aggregation over the view-wise consensus posteriors**. Each view first produces its **consensus posterior**, then a **normalized-entropy-based confidence** is computed for that posterior, these confidences are normalized into weights, and the weighted log-posteriors are aggregated to obtain the final posterior used for prediction. Thus, when multiple views are observed, the reported results come from **cross-view confidence-weighted consensus**, rather than from one privileged path or one fixed visible view. We now make this default inference rule explicit in the experimental setup section.
>
> We thank Reviewer `aqnR` again for the strong support and for recognizing both the novelty and the empirical strength of the paper. We hope the revised clarifications make the contribution, implementation details, and inference procedure much clearer.

---

> > ### Author Rebuttal · Reviewer_aqnR · 2026-04-02
> >
> > The authors provided a detailed response to the main concerns, which has solved all my concerns. Therefore, I'm maintaining my original score and recommend a strong accept.

---

### Official Review · Reviewer_2EVB · 2026-03-08

**Soundness:** 3
**Presentation:** 3
**Significance:** 3
**Originality:** 3
**Overall Recommendation:** 5
**Confidence:** 4

**Summary:**

This paper investigates two recurring failure modes in incomplete multi-view learning under varying missing rates: (i) scarce co-observed samples weaken cross-view alignment and may induce dominant-view bias; and (ii) overly early or overly aggressive imputation can magnify pseudo-label uncertainty and propagate errors. To address these issues, the authors propose MAGIC, which combines multi-path contrastive consensus with confidence-gated semantic imputation. The consensus module enforces consistency at both the representation and prediction levels across fused, per-view, and masked-fusion paths, thereby reducing reliance on scarce paired observations. The imputation module performs conservative class-level semantic transfer and is activated only when sufficient evidence is available through confidence gating, which helps limit unreliable error propagation. Overall, MAGIC improves robustness under high missingness and alleviates error accumulation caused by premature or noisy imputation.

**Compliance With Llm Reviewing Policy:**

Affirmed.

**Final Justification:**

My concerns have been addressed, and I would like to increase my rating.

**Key Questions For Authors:**

- For clarity and reproducibility, when only a subset of views is available at test time, what posterior is used by default to produce the final cluster label?

- To ensure fair comparison, do all methods use exactly the same missing masks for each dataset and missing rate $\eta$? How is missingness generated in practice: by independent Bernoulli sampling per view, or by view-wise visibility probabilities to simulate view imbalance? If the latter, are the masks shared across all compared methods?

- In Sec. 3.3 (Eq. (5)), could the authors specify the mixing coefficients for the fused, per-view, and masked-fusion paths, and clarify whether these coefficients are normalized? Are these coefficients fixed across datasets, or tuned separately for each dataset?

I will increase my rating if the authors could address my concerns&questions

**Limitations:**

Yes

**Strengths And Weaknesses:**

- The paper addresses weakened alignment signals and dominant-view bias under scarce co-observed samples by replacing single-path alignment with multi-path learning at both the representation and prediction levels. Cross-path consensus is enforced through a contrastive objective and a prediction-consistency regularizer, which improves stability and robustness under severe missingness.

- The method constructs a consensus posterior through a power-product-style aggregation of predictions across multiple paths, rather than treating any single path as a hard pseudo ground truth. This design reduces contamination from isolated but overconfident predictions, yielding better-calibrated soft targets that provide a stronger supervisory signal for subsequent semantic transfer and imputation.

- The imputation strategy follows a conservative, evidence-driven principle. Imputation is activated only after semantic calibration, while confidence gating regulates class-level label-distribution transfer. In addition, the prototype-based fallback at the feature level helps mitigate noise injection when neighborhood evidence is unreliable, thereby reducing uncertainty amplification and error accumulation.

- Across multiple benchmarks and missing rates, the method achieves the best overall performance among the evaluated approaches and degrades more gracefully as missingness increases. The ablation studies, sensitivity analyses, and convergence results help attribute the performance gains and suggest that the proposed modules provide complementary rather than redundant benefits.


Weaknesses
- Reproducibility could be further improved. Although the overall pipeline and notation are presented clearly, several important default settings are not explicitly specified in the main experimental setup, such as the imputation start epoch $T_{\text{imp}}$, whether the gating threshold $\tau_{\text{src}}$ varies during training, and the default loss weights and temperature parameters. A compact table summarizing default values and recommended ranges would make reproduction easier.

- The authors argue that the scarcity of co-observed cross-view pairs adversely affects MvC performance. However, I note that some recent studies (e.g., Semantic Invariant Multi-View Clustering With Fully Incomplete Information, Community-aware Multi-view Representation Learning with Incomplete Information) have shown that MvC can still be achieved even without co-observed pairs. It would strengthen the paper if the authors could clarify the relationship between their work and these studies, and more explicitly articulate the differences. This would help make the motivation and contribution of the paper more solid.

- The default inference rule is not fully clear. When some views are missing at test time, it remains ambiguous whether the final prediction is obtained from the fused posterior, a cross-view consensus posterior, or an aggregation over the visible views. A brief clarification of the default inference strategy used for all reported results would remove this ambiguity.

- There are minor writing and formatting issues, such as the duplicated dataset name in a figure caption (“Fashion and Fashion”). A careful proofreading pass would further improve the presentation quality.

---

> ### Author Rebuttal · Authors · 2026-03-30
>
> We thank Reviewer `2EVB` for the careful reading and constructive suggestions. We especially appreciate the reviewer’s recognition that **MAGIC addresses high-missingness clustering through multi-path consensus learning and confidence-gated semantic imputation**. We also thank the reviewer for noting that the ablations, sensitivity analyses, and convergence results support the effectiveness and stability of the method. We respond to the reviewer’s questions and concerns below.
>
> **W1.** Reproducibility could be improved because key defaults in the experimental setup are not fully explicit, including the imputation start epoch, whether the gating threshold changes, and the default loss weights and temperature parameters.
>
> **Respond:** These defaults are **fixed**. Training uses **100 epochs** of reconstruction pretraining and **100 epochs** of contrastive fine-tuning. Imputation is **disabled for the first 30 epochs**, **OT-based semantic imputation starts at epoch 30**, and **KNN refinement starts at epoch 50**. The semantic-transfer threshold is **dynamic**, decreasing from **0.65 to 0.55**. The label-level contrastive temperature is **0.7 by default** and **0.5 on Fashion**, and normalization is predetermined per dataset. Posterior aggregation uses **fixed mixing weights** for the fused, per-view, and masked-fusion paths with a **fixed aggregation temperature of 0.8**. The masking branch uses **feature-dimension-level random masking** with ratio **0.7**, **six masked-fusion samples**, and view-drop probability **0.35**. Prototypes are **recomputed online in each forward pass** from the current soft assignments and visible samples, without a memory bank or momentum update. We have added a compact implementation summary in the revised experimental setup.
>
> **W2.** The default inference rule is not fully clear. When views are missing at test time, it is ambiguous whether the final prediction comes from the fused posterior, a cross-view consensus posterior, or an aggregation over the visible views.
>
> **Respond:** The default inference rule is **not** the raw fused posterior. At test time, the model first computes a **view-wise consensus posterior for each view**, then forms the final prediction by a **confidence-weighted aggregation** of these posteriors. Specifically, we compute a **normalized-entropy-based confidence** for each view posterior, normalize these confidences into weights, aggregate the weighted log-posteriors, and take the argmax. Thus, when views are missing, the reported results come from **confidence-weighted cross-view consensus**. We have made this rule explicit in the revised experimental setup.
>
> **Q1.** For clarity and reproducibility, when only a subset of views is available at test time, what posterior is used by default to produce the final cluster label?
>
> **Respond:** The final cluster label is produced from the **confidence-weighted aggregation of the view-wise consensus posteriors**, **not** from a single-view posterior or the raw fused posterior.
>
> **Q2.** To ensure fair comparison, do all methods use exactly the same missing masks for each dataset and missing rate? How is missingness generated in practice?
>
> **Respond: Yes.** For each dataset and missing-rate setting, the **same missing mask is generated once and reused across methods**. Missingness is **not** generated by independent Bernoulli sampling per view, and it is **not** based on manually specified visibility probabilities. Instead, we use a **constrained random procedure**: some samples are kept fully observed; for the rest, we ensure that **each sample retains at least one observed view** by first assigning one preserved view through one-hot sampling, then adding extra observed entries by random sampling so that the final observed-view ratio matches the target missing rate as closely as possible. The rows are then randomly permuted to form the final mask matrix. Because the mask is generated once and saved for each dataset and missing-rate setting, it is **shared across all compared methods**.
>
> **Q3.** In Sec. 3.3 (Eq. (5)), could the authors specify the mixing coefficients for the fused, per-view, and masked-fusion paths, and clarify whether these coefficients are normalized? Are these coefficients fixed across datasets, or tuned separately?
>
> **Respond:** In Eq. (5), the mixing coefficients are implemented through **fixed weights** for the fused, per-view, and masked-fusion paths and are **implicitly normalized** by dividing by their sum in the barycentric log-opinion aggregation. In the default setting, the effective normalized coefficients are **1/3**, **1/6**, and **1/2**, and the aggregation temperature is **$0.8$**. These settings are **fixed in the main experiments** rather than tuned separately for each dataset.
>
> We thank Reviewer `2EVB` again for the helpful suggestions. We hope these clarifications make the default settings, inference rule, mask-generation protocol, and aggregation details much clearer.

---

> > ### Author Rebuttal · Reviewer_2EVB · 2026-04-01
> >
> > My concerns have been addressed, and I would like to increase my rating.

---

### Official Review · Reviewer_wG4Y · 2026-03-10

**Soundness:** 3
**Presentation:** 4
**Significance:** 3
**Originality:** 3
**Overall Recommendation:** 5
**Confidence:** 5

**Summary:**

This paper presents MAGIC for robust incomplete multi-view clustering under high missingness and biased view visibility, and the evidence is strong enough to merit a strong accept. (1) It introduces a two-stage training recipe that first stabilizes semantic posteriors via multi-path consensus learning, reducing alignment fragility when co-observed pairs are scarce. (2) It then performs controlled semantic propagation using normalized-entropy confidence gating, enabling class-level transfer only under sufficient reliability. (3) A prototype fallback further constrains missing-view representations and limits error diffusion, and the extensive experiments with ablations and stability analyses support both effectiveness and trainability.

**Compliance With Llm Reviewing Policy:**

Affirmed.

**Ethics Expertise Needed:**

["Other Expertise"]

**Final Justification:**

All of my concerns have been addressed, and I will keep my score.

**Key Questions For Authors:**

(1). Please clarify whether ACC is computed with Hungarian matching (or another label-permutation alignment), which library/version is used for NMI/ARI, and whether the same evaluation code path is applied to all methods.
(2). Please report the default gating threshold used in the main results (and whether it changes over training), and specify when imputation is enabled (e.g., starting epoch or a triggering condition).
(3). For the t-SNE plots, which representation is used (a specific view embedding, fused embedding, or the aggregated representation), and are key t-SNE hyperparameters (e.g., perplexity and number of iterations) fixed across methods/settings?

**Limitations:**

Yes

**Strengths And Weaknesses:**

Strengths:
(1). The paper avoids treating imputation as a default prerequisite. It first stabilizes a trustworthy semantic posterior, then propagates semantics cautiously via gating. This “stabilize first, then impute” order naturally reduces error cascades under high missingness in unsupervised settings.
(2). The design cleanly splits high-missingness risks into two stages. Multi-path learning extracts stable shared semantics under scarce co-observations and imbalanced visibility, while confidence gating plus prototype constraints keep transfer and imputation conservative and controllable.
(3). Results span multiple missing rates and remain competitive in the hardest regimes. Ablations, convergence curves, and sensitivity analyses go beyond reporting numbers and provide credible evidence for stability and trainability.
(4). Despite multiple modules, the exposition is generally smooth. The formulas and algorithm description are sufficiently detailed for readers to reconstruct the implementation with modest effort, supporting reproducibility.

Weaknesses:
(1). Several implementation details would benefit from being fixed as defaults, including the exact perturbations used in each multi-path branch, the mask granularity/ratio in masked-fusion, temperature/normalization choices, and the prototype update rule.
(2). A final pass on presentation is still needed to remove any residual template text, ensure consistent notation, and eliminate minor typos in captions/terms, so the write-up matches the strong-accept technical quality.

---

> ### Author Rebuttal · Authors · 2026-03-30
>
> We thank Reviewer `wG4Y` for the positive assessment and for clearly recognizing the core contribution of our paper: **MAGIC is a two-stage “stabilize first, then impute” framework** for incomplete multi-view clustering under high missingness and biased view visibility. In particular, the reviewer accurately captured that our method first improves posterior reliability through **multi-path consensus learning**, and then performs **controlled semantic propagation and imputation** through confidence gating and prototype constraints. We also appreciate the reviewer’s recognition that the ablations, convergence curves, and sensitivity analyses provide credible evidence for both effectiveness and trainability, and that the overall exposition is sufficiently clear and detailed for reproducibility. To address the reviewer’s questions and remaining concerns, we respond point by point below.
>
> **Q1.** Please clarify whether ACC is computed with Hungarian matching (or another label-permutation alignment), which library/version is used for NMI/ARI, and whether the same evaluation code path is applied to all methods.
>
> **Respond:** **Yes.** ACC is computed after **optimal label-permutation alignment** using the **Hungarian assignment algorithm**. NMI and ARI are computed using the **standard scikit-learn implementations**. Importantly, **all reported results go through the same evaluation pipeline**: both per-view cluster assignments and final semantic labels are evaluated with the **same metric routine**. In the manuscript revision we have made this evaluation protocol explicit, so the contribution is supported by a fully consistent comparison setup.
>
> **Q2.** Please report the default gating threshold used in the main results (and whether it changes over training), and specify when imputation is enabled (e.g., starting epoch or a triggering condition).
>
> **Respond:** It is important to distinguish **two different quantities**. The **aggregation temperature** in the barycentric combination of the three path-wise posteriors is **fixed at 0.8**. By contrast, the actual **confidence threshold for semantic transfer**, denoted by **$\tau_{\mathrm{src}}$**, is **dynamic rather than fixed**. During contrastive training, imputation is **disabled for the first 30 epochs**, **OT-based semantic imputation starts at epoch 30**, and **kNN refinement starts at epoch 50**. Meanwhile, **$\tau_{\mathrm{src}}$ decreases linearly from 0.65 to 0.55**, so the gate is stricter when semantic transfer is first activated and becomes slightly more permissive only after the posteriors become more stable. We have already made this schedule explicit in the revised implementation summary.
>
> **Q3.** For the t-SNE plots, which representation is used (a specific view embedding, fused embedding, or the aggregated representation), and are key t-SNE hyperparameters (e.g., perplexity and number of iterations) fixed across methods/settings?
>
> **Respond:** Our t-SNE visualizations use the **fused representation**, rather than the embedding of any single view. This choice is intended to reflect the **final cross-view semantic structure** learned by the model, rather than a view-specific latent space. For large datasets, we follow a **unified preprocessing rule**: when the number of samples exceeds **5000**, we first apply **PCA to reduce the representation to 50 dimensions** and then run t-SNE; otherwise, t-SNE is applied directly to the fused representation. Thus, the representation choice and preprocessing pipeline are **fixed by a shared visualization script** rather than tuned separately for different methods or settings. We have already revised the corresponding description so that the manuscript explicitly states the visualization input, preprocessing rule, and fixed usage protocol.
>
> We thank Reviewer `wG4Y` again for the strong support and helpful questions. We hope the revised clarifications make the method contribution, default settings, evaluation protocol, and visualization protocol much clearer.

---

> > ### Author Rebuttal · Reviewer_wG4Y · 2026-04-01
> >
> > All of my concerns have been addressed, and I will keep my score.

---

### Review · Ethics_Reviewer_WbC8 · 2026-03-28

**Recommendation:** No remediation action needed

**Ethics Issue:**

I can see no issues with this work. Only one reviewer raised a flag, and they didn't elaborate on their concern in their review.

---

### Decision · Program_Chairs · 2026-04-30

**Decision:**

Accept (regular)

**Comment:**

This paper studies incomplete multi-view clustering under high missingness and imbalanced view visibility, and proposes the MAGIC framework. A major strength of the paper is that it accurately identifies two central challenges in this setting, namely fragile cross-view alignment caused by scarce co-observed samples and error accumulation caused by premature semantic propagation and imputation, and addresses them through a structurally clear and logically coherent two-stage solution.

 The paper’s central idea of first stabilizing semantic posteriors and then performing controlled propagation and imputation was positively received by Reviewers wG4Y, 2EVB, aqnR, and uYjd. The initial scores from these reviewers were 5, 4, 6, and 5, respectively; after the rebuttal, the final scores were 5, 5, 6, and 5, and all four reviewers stated that their main concerns had been adequately addressed.

Overall, based on the reviews, the author response, and the overall discussion, I recommend acceptance. The authors should conduct a careful proofreading pass, improve a small number of textual descriptions, and correct minor issues in several figures and figure captions in their final version.